# Coherent spin qubit shuttling through germanium quantum dots

Floor van Riggelen-Doelman [1], Chien-An Wang [1], Sander L. de Snoo [1], William I. L. Lawrie [1], Nico W. Hendrickx[1], Maximilian Rimbach-Russ [1], Amir Sammak[2], Giordano Scappucci [1], Corentin Déprez [1] ✉ & Menno Veldhorst [1] ✉

Quantum links can interconnect qubit registers and are therefore essential in networked quantum computing. Semiconductor quantum dot qubits have seen significant progress in the high-fidelity operation of small qubit registers but establishing a compelling quantum link remains a challenge. Here, we show that a spin qubit can be shuttled through multiple quantum dots while preserving its quantum information. Remarkably, we achieve these results using hole spin qubits in germanium, despite the presence of strong spin-orbit interaction. In a minimal quantum dot chain, we accomplish the shuttling of spin basis states over effective lengths beyond 300 microns and demonstrate the coherent shuttling of superposition states over effective lengths corresponding to 9 microns, which we can extend to 49 microns by incorporating dynamical decoupling. These findings indicate qubit shuttling as an effective approach to route qubits within registers and to establish quantum links between registers.

The envisioned approach for semiconductor spin qubits towards fault-tolerant quantum computation centers on the concept of quantum networks, where qubit registers are interconnected via quantum links[1]. Significant progress has been made in controlling few-qubit registers[2,3]. Recent efforts have led to demonstrations of high fidelity single- and two-qubit gates[4–7], quantum logic above one Kelvin[8–10] and operation of a 16 quantum dot array[11]. However, scaling up to larger qubit numbers requires changes in the device architecture[12,13].

Inclusion of short-range and mid-range quantum links could be particularly effective to establish scalability, addressability, and qubit connectivity. The coherent shuttling of electron or hole spins is an appealing concept for the integration of such quantum links in spin qubit devices. Short-range coupling, implemented by shuttling a spin qubit through quantum dots in an array, can provide flexible qubit routing and local addressability[14,15]. Moreover, it allows to increase connectivity beyond nearest-neighbor coupling and decrease the number of gates needed to execute algorithms. Mid-range links, implemented by shuttling spins through a multitude of quantum dots,

may entangle distant qubit registers for networked computing and allow for qubit operations at dedicated locations[14,16–18]. Furthermore, such quantum buses could provide space for the integration of on-chip control electronics[1], depending on their footprint.

The potential of shuttling-based quantum buses has stimulated research on shuttling electron charge[19–21] and spin[15,22–29]. While nuclear spin noise prevents high-fidelity qubit operation in gallium arsenide, demonstrations of coherent transfer of individual electron spins through quantum dots are encouraging[22–26]. In silicon, qubits can be operated with high-fidelity and this has been employed to displace a spin qubit in a double quantum dot[15,27]. Networked quantum computers, however, will require integration of qubit control and shuttling through chains of quantum dots, incorporating quantum dots that have at least two neighbors.

Meanwhile, quantum dots defined in strained germanium (Ge/SiGe) heterostructures have emerged as a promising platform for hole spin qubits[30,31]. The high quality of the platform allowed for rapid development of single spin qubits[32,33], singlet-triplet qubits[34–36], a four

[1]QuTech and Kavli Institute of Nanoscience, Delft University of Technology, PO Box 5046, 2600 GA Delft, The Netherlands. [2]QuTech and Netherlands Organisation for Applied Scientific Research (TNO), 2628 CK Delft, The Netherlands. ✉e-mail: c.c.deprez@tudelft.nl; m.veldhorst@tudelft.nl

qubit processor[2], and a $4 \times 4$ quantum dot array with shared gate control[11]. While the strong spin-orbit interaction allows for fast and all-electrical control, the resulting anisotropic $g$-tensor[31,37] complicates the spin dynamics and may challenge the feasibility of a quantum bus.

Here, we demonstrate that spin qubits can be shuttled through quantum dots. These experiments are performed with two hole spin qubits in a $2 \times 2$ germanium quantum dot array. Importantly, we operate in a regime where we can implement single-qubit logic and coherently transfer spin qubits through an intermediate quantum dot. Furthermore, by performing experiments with precise voltage pulses and sub-nanosecond time resolution, we can mitigate finite qubit rotations induced by spin-orbit interactions. In these optimized sequences we find that the shuttling performance is limited by dephasing and can be extended through dynamical decoupling.

## Results
### Coherent shuttling of single hole spin qubits

Figure 1a shows a germanium $2 \times 2$ quantum dot array identical to the one used in the experiment[2]. The chemical potentials and the tunnel couplings of the quantum dots are controlled with virtual gates (vP$_i$, vB$_{ij}$), which consist of combinations of voltages on the plunger gates and the barrier gates. We operate the device with two spin qubits in quantum dots QD$_1$ and QD$_2$ initialized in the $|\downarrow\downarrow\rangle$ state (see Methods). We use the qubit in QD$_1$ as an ancilla to readout the hole spin in QD$_2$, using latched Pauli spin blockade[2,38,39]. The other qubit starts in QD$_2$ and is shuttled to the other quantum dots by changing the detuning energies ($\epsilon_{23/34}$) between the quantum dots (Fig. 1b, e, i). The detuning energies are varied by pulsing the plunger gate voltages as illustrated in Fig. 1f, j. Additionally, we increase the tunnel couplings between QD$_2$-QD$_3$ and QD$_3$-QD$_4$ before shuttling to allow for adiabatic charge transfer. The hole carrying the spin remains in its orbital ground state and, with increasing $|\epsilon|$, the charge becomes localized in the quantum dot with the lowest chemical potential as displayed in Fig. 1b. In our experiments, we tune to have adiabatic evolution with respect to charge, and study adiabatic and diabatic shuttling with respect to spin.

The $g$-tensor of hole spin qubits in germanium is sensitive to the local electric field. Therefore, the Larmor frequency $f_L$ is different in

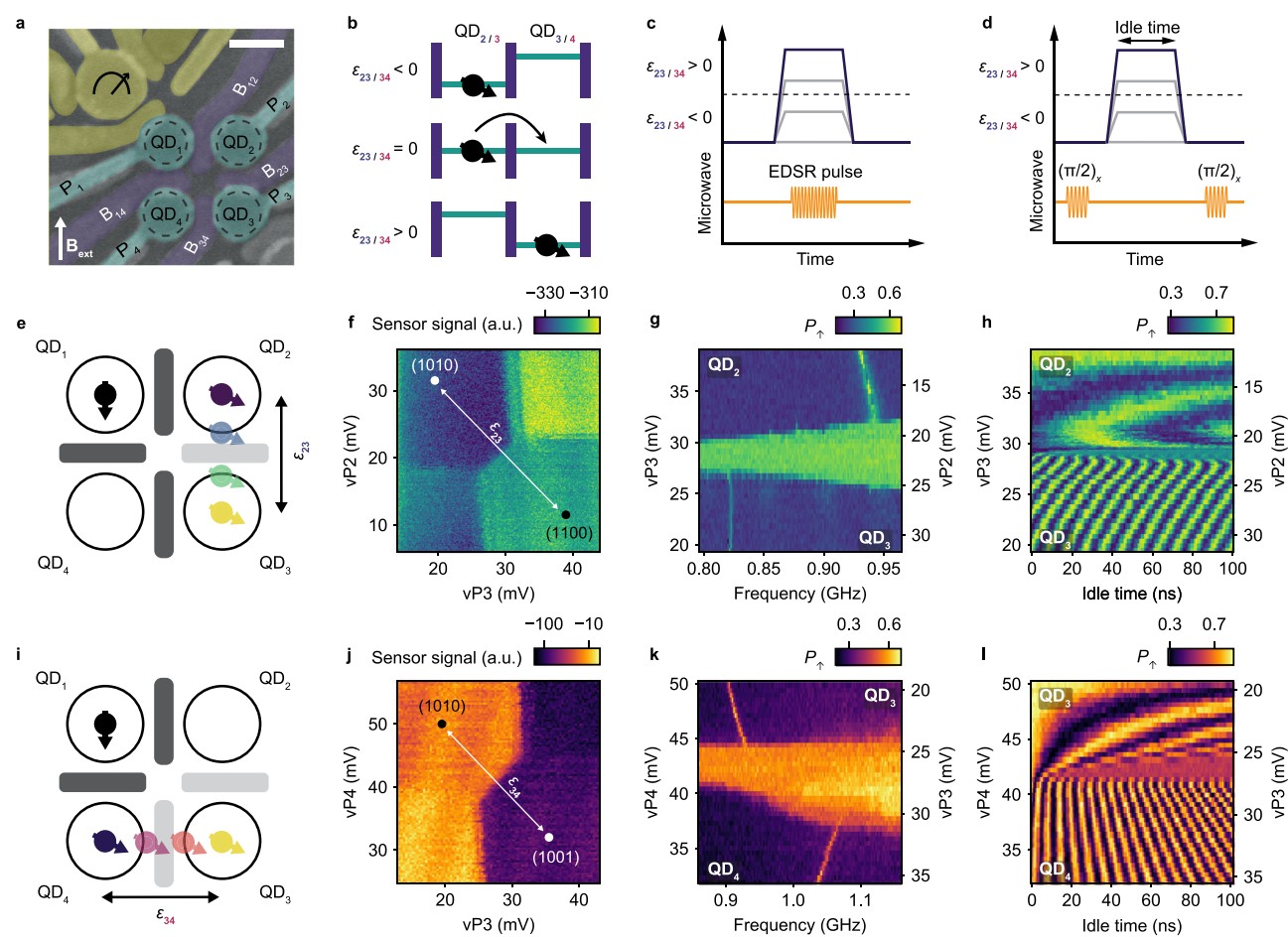

**Fig. 1 | Coherent shuttling of hole spin qubits in germanium double quantum dots. a** False colored scanning electron microscope image of a representative quantum dot (QD) device. The scale bar corresponds to 100 nm. Unless specified otherwise, an external magnetic field of 0.25 T is applied in the direction indicated by the arrow. **b** Schematic showing the principle of bucket-brigade-mode shuttling. The detuning energy $\epsilon_{23/34}$ between the two quantum dots is progressively changed such that it becomes energetically favorable for the hole to tunnel from one quantum dot to another. **c** Pulse sequence for experiments shown in **g** and **k**. EDSR stands for Electric Dipole Spin Resonance. **d** Pulse sequence for coherent shuttling shown in **h** and **l**. **e, i** Schematics illustrating the direction of spin qubit shuttling investigated in **f**–**h** and **j**–**l** respectively. Charge stability diagrams of QD$_2$-QD$_3$ (**f**) and QD$_3$-QD$_4$ (**j**). To shuttle the qubit from one site to another, the virtual plunger gate voltages are varied along the detuning axis (white arrow), which crosses the interdot charge transition line. The labels ($N_1N_2N_3N_4$) represent the charge occupation in the quantum dots. Probing of the resonance frequency along the detuning axis for the double quantum dot QD$_2$-QD$_3$ (**g**) and QD$_3$-QD$_4$ (**k**). $P_\uparrow$ (the probability of measuring the qubit in the $|\uparrow\rangle$ state) is obtained at the end of the pulse sequence. The duration of the microwave pulse is 4 µs. The ramp time used to change the detuning is 40 ns for the measurement shown in **g** and 12 ns for the measurement shown in **k**. Nearby the charge transition, the resonance frequency cannot be resolved due to a combination of effects discussed in Supplementary Note 1. Shuttling of superposition states between QD$_2$-QD$_3$ (**h**) and QD$_3$-QD$_4$ (**l**). The ramp time used to change the detuning is 40 ns for the measurement shown in **h** and 4 ns for the measurement shown in **l**.

each quantum dot[32–34]. We exploit this effect to confirm the shuttling of a hole spin from one quantum dot to another. In Fig. 1c. we show the experimental sequence used to measure the qubit resonance frequency, while changing the detuning to transfer the qubit. Figure 1g (k) shows the experimental results for spin transfers from $QD_2$ to $QD_3$ ($QD_3$ to $QD_4$). Two regions can be clearly distinguished in between which $f_L$ varies by 110 (130) MHz. This obvious change in $f_L$ clearly shows that the hole is shuttled from $QD_2$ to $QD_3$ ($QD_3$ to $QD_4$) when applying a sufficiently large detuning pulse. To investigate whether such transfer is coherent, we probe the free evolution of qubits prepared in a superposition state after applying a detuning pulse (Fig. 1d)[27]. The resulting coherent oscillations are shown in Fig. 1h (l). They are visible over the full range of voltages spanned by the experiment and arise from a phase accumulation during the idle time. Their frequency $f_{osc}$ is determined by the difference in resonance frequency between the starting and the end points in detuning as shown in Supplementary Fig. 1. The abrupt change in $f_{osc}$ marks the point where the voltage pulse is sufficiently large to transfer the qubit from $QD_2$ to $QD_3$ ($QD_3$ to $QD_4$). These results clearly demonstrate that single hole spin qubits can be coherently transferred.

### The effect of strong spin-orbit interaction on spin shuttling

The strong spin-orbit interaction in our system has a significant impact on the spin dynamics during the shuttling. It appears when shuttling a qubit in a $|\downarrow\rangle$ state between $QD_2$ and $QD_3$ using fast detuning pulses with voltage ramps of 4 ns. Doing this generates coherent oscillations shown in Fig. 2b that appear only when the qubit is in $QD_3$. They result from the strong spin-orbit interaction and the use of an almost in-plane magnetic field[40]. In this configuration, the direction of the spin quantization axis depends strongly on the local electric field[35,37,41–43] and can change significantly between neighboring quantum dots. Therefore, rapid shuttling of a hole results in a change of angle between the spin state and the local spin quantization axis. In particular, a qubit in a basis state in $QD_2$

becomes a qubit in a superposition state in $QD_3$ when it is shuttled diabatically with respect to the change in quantization axis. Consequently, the spin precesses around the quantization axis of $QD_3$ until it is shuttled back (Fig. 2a). This leads to qubit rotations and the aforementioned oscillations.

While these oscillations are clearly visible for voltage pulses with ramp times $t_{ramp}$ of few nanoseconds, they fade as the ramp times are increased, as shown in Fig. 2c, and vanish for $t_{ramp} > 30$ ns. The qubit is then transferred adiabatically, can follow the change in quantization axis and therefore remains in the spin basis state in both quantum dots. From the visibility of the oscillations, we estimate that the quantization axis of $QD_3$ ($QD_4$) is tilted by at least 42° (33°) compared to the quantization axis of $QD_2$ ($QD_3$). These values are corroborated by independent estimations made by fitting the evolution of $f_L$ along the detuning axes (see Supplementary Note 2).

Figure 2d, e displays the magnetic field dependence of the oscillations generated by diabatic shuttling. Their frequencies $f_{osc}$ increase linearly with the field and match the Larmor frequencies $f_L$ measured for a spin in the target quantum dot. This is consistent with the explanation that the oscillations are due to the spin precessing around the quantization axis of the second quantum dot.

### Shuttling performance

To quantify the performance of shuttling a spin qubit, we implement the experiments depicted in Fig. 3a, e, f[15,27] and study how the state of a qubit evolves depending on the number of subsequent shuttling events. For hole spins in germanium, it is important to account for rotations induced by the spin-orbit interaction. This can be done by aiming to avoid unintended rotations, or by developing methods to correct them. An example of the first approach is transferring the spin qubits adiabatically. This implies using voltage pulses with ramps of tens of nanoseconds, which are significant with respect to the dephasing time. However, this strongly limits the shuttling performance (see Supplementary Fig. 9). Instead, we can mitigate rotations

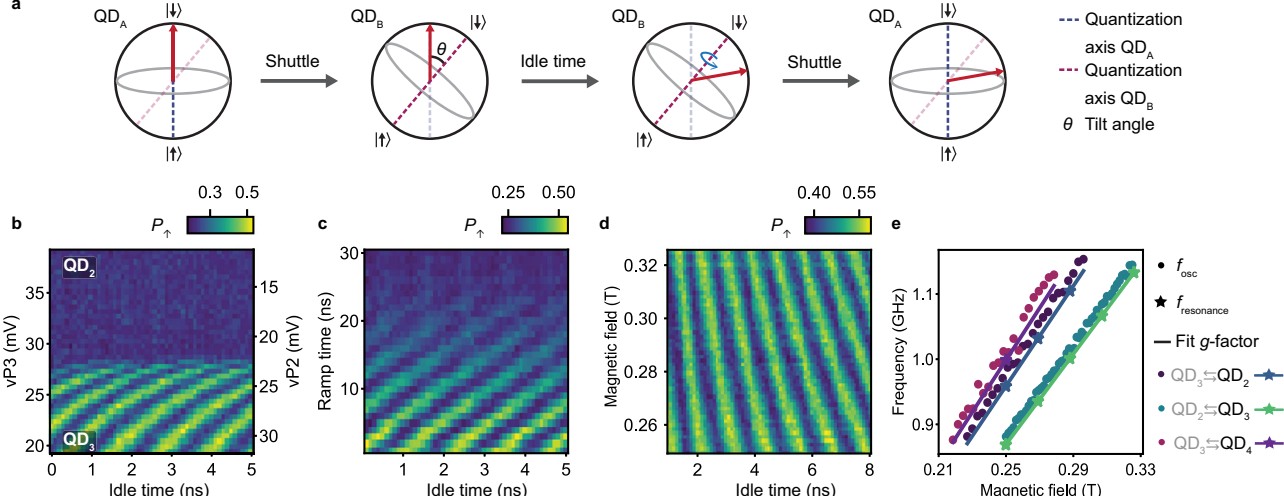

**Fig. 2 | Rotations induced while shuttling by the difference in quantization axes. a** Schematic explaining the effect of the change in quantization axis direction that the qubit experiences during the shuttling process. The difference in quantization axis between quantum dots (QDs) is caused by the strong spin-orbit interaction. **b** Oscillations in spin-up probabilities $P_\uparrow$ induced by the change in quantization axis while shuttling diabatically a qubit in a $|\downarrow\rangle$ state between $QD_2$ and $QD_3$. Ramp times of 4 ns are used for the detuning pulses. Note that the oscillations have a reduced visibility, meaning that the difference in quantization axes does not induce a full spin flip. The angle between the quantization axes of the two quantum dots can be estimated from the amplitude of the oscillations, see Supplementary Note 2A. **c** Oscillations due to the change in quantization axis at a fixed point in

detuning, as function of the voltage pulse ramp time used to shuttle the spin. When the ramp time is long enough, typically above 30 ns, the spin is shuttled adiabatically and the oscillations vanish. **d** Magnetic-field dependence of the oscillations induced by the difference in quantization axis. **e** Frequency of the oscillations $f_{osc}$ induced by the change in quantization axis as a function of magnetic field for different shuttling processes. The oscillation frequency $f_{osc}$ for $QD_3$ is extracted from measurements displayed in **d** (and similar experiments for the other quantum dot pairs) and is plotted with points. $f_{osc}$ scales linearly with the magnetic field. Comparing $f_{osc}$ with resonance frequencies $f_{resonance}$ measured using microwave pulses (data points depicted with stars) reveals that $f_{osc}$ is given by the Larmor frequency of the quantum dot towards which the qubit is shuttled (black label).

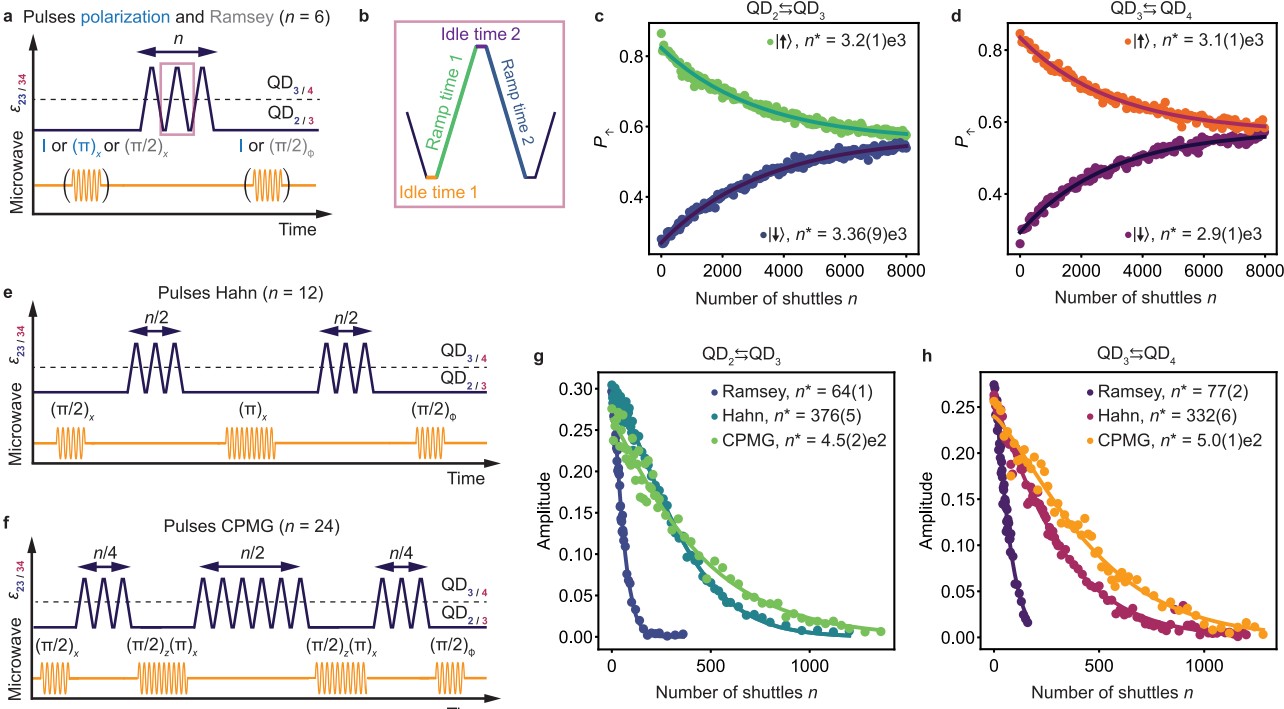

**Fig. 3 | Quantifying the performance for the shuttling in double quantum dots.**
**a** Schematic of the pulse sequence used for quantifying the performance of shuttling basis states (blue) or a superposition state (gray). The spin qubit is prepared in the quantum dot (QD) where the shuttling experiment starts, by either applying an identity gate (shuttling a $|\downarrow\rangle$ state), a $(\pi)_x$ pulse (shuttling a $|\uparrow\rangle$ state) or $(\pi/2)_x$ pulse (shuttling a superposition state referred as Ramsey shuttling experiments). Detuning pulses are applied to the plunger gates to shuttle the hole from one QD to another, back and forth, and finally the appropriate pulses are applied to prepare for readout. Moving the qubit from one QD to another is counted as one shuttle $n = 1$. Since the hole is always shuttled back for readout, $n$ is always even. The schematic shows an example for $n = 6$. **b** Zoom-in on the detuning pulses used for the shuttling. To make an integer number of $2\pi$ rotation(s) around the quantization axis of the second QD, all ramp and idle times in the pulse need to

be optimized. Spin-up probabilities $P_\uparrow$ measured after shuttling $n$ times a qubit prepared in a spin basis state between QD$_2$ and QD$_3$ (**c**) and between QD$_3$ and QD$_4$ (**d**). The decays of $P_\uparrow$ are fitted (solid lines) to an exponential function $P_\uparrow = P_0 \exp(-n/n^*) + P_{\text{sat}}$. **e** Pulse sequence used for implementing a Hahn echo shuttling experiment. In the middle of the shuttling experiment, an echo pulse $(\pi)_x$ is applied in the QD where the spin qubit was initially prepared. Example for $n = 12$. **f** Pulse sequence for a CPMG shuttling experiment. Two $(\pi/2)_z(\pi)_x$ pulses are inserted between the shuttling pulses. Example for $n = 24$. Performance of the shuttling of superposition state between QD$_2$ and QD$_3$ (**g**) and QD$_3$ and QD$_4$ (**h**) for different shuttling sequences. The decays of the coherent amplitude $A$ of the superposition state are fitted (solid lines) by $A_0 \exp(-(n/n^*)^\alpha)$ where $\alpha$ is a fitting parameter. The uncertainties indicate one standard deviation from the best fits.

by carefully tuning the duration of the voltage pulses, such that the qubit performs an integer number of $2\pi$ rotations around the quantization axis of the respective quantum dot. This approach is demanding, as it involves careful optimization of the idle times in each quantum dot as well as the ramp times, as depicted in Fig. 3b. However, it allows for fast shuttling, with ramp times of typically 4 ns and idle times of 1 ns, significantly reducing the dephasing experienced by the qubit during the shuttling. We employ this strategy in the rest of our experiments.

We first characterize the shuttling of a spin qubit initialized in a basis state. We do this by preparing a qubit in a $|\uparrow\rangle$ or $|\downarrow\rangle$ state and transferring it multiple times between the quantum dots. Figure 3c, d displays the spin-up fraction $P_\uparrow$ measured as a function of the number of shuttling steps $n$. The probability of ending up in the initial state shows a clear exponential dependence on $n$. No oscillations of $P_\uparrow$ with $n$ are visible, confirming that the pulses have been successfully optimized to account for unwanted spin rotations. We extract the characteristic decay constants $n^*$ by fitting the data for the shuttling of qubits prepared in $|\uparrow\rangle$ and $|\downarrow\rangle$ states separately as they originate from distinct sets of experiments. In all cases, we find a characteristic decay $n^* \simeq 3000$ shuttles between quantum dots, corresponding to a polarization transfer fidelities of $F = \exp(-1/n^*) \simeq 99.97$ % per shuttle within the sequence. This is similar to the fidelities reached in silicon devices[15,27], despite the anisotropic $g$-tensors due to the strong spin-orbit interaction in our platform.

The exponential decay of the spin polarization to approximately 0.5 can emerge from different effects. At the charge anticrossing, the spin polarization life time is strongly reduced (see Supplementary Fig. 3), due to high frequency charge noise and coupling to phonons[44]. Passing the charge anticrossing repeatedly thus leads to a randomization of the spin. Moreover, while the qubit starts in a basis state, it undergoes coherent rotations due to the diabatic spin shuttling and it is in a superposition state in the second quantum dot. The qubit, although initially in a spin basis state, then becomes sensitive to dephasing which can also lead to an exponential decay of $P_\uparrow$. The experimental decay observed probably results from a combination of these mechanisms.

We emphasize that the exact impact of dephasing on the performance of the shuttling of spin basis state depends on the difference in quantization axes of the quantum dots and on the pulse sequence used (see Supplementary Note 8). In our experiment, the dephasing is probably mitigated by a decoupling effect induced by repeatedly waiting in the initial quantum dot (see explanations Supplementary Note 8). While extrapolating this result to a long chain of quantum dots is not straightforward, similar noise-averaging effects may occur in the presence of spatially correlated noise in the chain[45]. In the absence of decoupling effects and for the purpose of shuttling basis states, adiabatic shuttling still provides a good alternative as we find $n^*$ to remain above 1000, corresponding to fidelities per shuttle within the sequence above 99.90% (see Supplementary Fig. 9).

We now focus on the performance of coherent shuttling. We prepare a superposition state via an EDSR $(\pi/2)_x$ pulse, shuttle the qubit, apply another $\pi/2$ pulse and measure the spin state. Importantly, one must account for $\hat{z}$-rotations experienced by the qubits during the experiments and the corresponding phase accumulation defined with respect to the qubit rotating frame in the initial quantum dots. The latter can be equivalently defined with respect to the lab frame. Therefore, we vary the phase of the EDSR pulse $\phi$ for the second $\pi/2$ pulse i.e. the final pulse is a $(\phi)_z(\pi/2)_x = (\pi/2)_\phi$ pulse. For each $n$, we then extract the amplitude $A$ of the $P_\uparrow$ oscillations that appear as function of $\phi$[15,27]. Figure 3g, h shows the evolution of $A$ as a function of $n$ for shuttling between adjacent quantum dots. We fit the experimental results using $A_0 \exp(-(n/n^*)^\alpha)$ and find characteristic decay constants $n^*_{23} = 64 \pm 1$ and $n^*_{34} = 77 \pm 2$. Remarkably, these numbers compare favorably to $n^* \simeq 50$ measured in a SiMOS electron double quantum dot[27], where the spin-orbit coupling is weak.

The exponents, $\alpha_{23} = 1.36 \pm 0.05$ and $\alpha_{34} = 1.28 \pm 0.06$, characterize the spectrum of the noise experienced by the qubit while it is shuttled and suggest that the noise is neither purely quasistatic nor white. The non-integer values of $\alpha$ contrast with observations in silicon[15,27], and suggest that the shuttling of hole spins in germanium is limited by other mechanisms. Two types of errors can be distinguished. Errors may occur during the diabatic part of the spin dynamics. On the other hand, errors can also be induced by the dephasing experienced by the qubits during the finite time spent in each quantum dot, including the ramp times (see Supplementary Note 8). To investigate the effect of dephasing, we modify the shuttling sequence and include a $(\pi)_x$ echoing pulse in the middle as displayed in Fig. 3e. We note that the echoing pulses are defined with respect to the rotating frame of the qubit in the starting quantum dots. Figure 3g, h shows the experimental results and it is clear that in germanium the coherent shuttling performance is improved significantly using an echo pulse: we can extend the shuttling by a factor of four to five, reaching a characteristic decay of more than 300 shuttles. Similarly, the use of CPMG sequences incorporating two decoupling $(\pi/2)_z(\pi)_x$ pulses (Fig. 3f) allows further, although modest, improvements. These enhancements in the shuttling performance confirm that dephasing is limiting the shuttling performance, contrary to observations in SiMOS[27]. We speculate that the origin of the difference is two-fold. Firstly, due to the stronger spin-orbit interaction, the spin is more sensitive to charge noise, resulting in shorter dephasing times. Secondly, the excellent control over the potential landscape in germanium allows minimizing the errors which are due to the shuttling itself.

While the results obtained for the diabatic shuttling in germanium double quantum dots are similar to those attained in silicon devices for adiabatic shuttling[15,27], one should be careful in comparing and extrapolating them to predict the performance of shuttling through longer quantum dot chains. Quantum dot chains that would allow to couple spin qubits over appreciable length scales will put higher demands on tuning, on uniformity, and the ability to tune all couplings. Moreover, a qubit shuttled through a chain may probe different noise environments which can further affect the performance.

## Shuttling through intermediate quantum dots

For distant qubit coupling, it is essential that a qubit can be coherently shuttled through chains of quantum dots. This is more challenging, as it requires control and optimization of a larger amount of parameters while more noise sources may couple to the system. Within a chain, a quantum dot will have at least two neighbors. To transport spin states from one site to another they have to pass through intermediate quantum dots. Therefore, an array of three quantum dots could be considered as the minimum size to explore the performance of shuttling in a chain.

We perform two types of experiments to probe the shuttling through chains of quantum dots, labeled corner shuttling and triangular shuttling. Figure 4b shows a schematic of the corner shuttling, which consists of transferring a qubit from $QD_2$ to $QD_3$ to $QD_4$ and back along the same route. The triangular shuttling, depicted in Fig. 4e, consists of shuttling the qubit from $QD_2$ to $QD_3$ to $QD_4$, and then directly back to $QD_2$, without passing through $QD_3$ (for the charge stability diagram $QD_4$-$QD_2$ and a detailed description see Supplementary Note 5).

To probe the feasibility of shuttling through a quantum dot, we first measure the free evolution of a superposition state while varying the detuning between the respective quantum dots. The results are shown in Fig. 4a. We find a remarkably clear coherent evolution for hole spin transfer from $QD_2$ to $QD_3$ to $QD_4$ and to $QD_2$. We observe one sharp change in the oscillation frequency for each transfer to the next quantum dot. We also note that after completing one round of the triangular shuttling, the phase evolution becomes constant, in agreement with a qubit returning to its original position. We thereby conclude that we can shuttle through quantum dots as desired.

We now focus on quantifying the performance of shuttling through quantum dots by repeated shuttling experiments. To allow comparisons with previous experiments, we define $n$ as the number of shuttling steps between two quantum dots. Meaning that one cycle in the corner shuttling experiments results in $n = 4$, while a loop in the triangular shuttling takes $n = 3$ steps. The results for shuttling basis states are shown in Fig. 4c, f. We note that the spin polarization decays faster compared to the shuttling in double quantum dots, in particular for the triangular shuttling. The corresponding fidelities per shuttle within the sequence are $F \simeq 99.96\%$ for the corner shuttling and $F \geq 99.63\%$ for the triangular shuttling.

For the corner shuttling, the faster decay of the basis states suggests a slight increase of the systematic error per shuttle. This may originate from the use of a more elaborated pulse sequence, which makes pulse optimization more challenging. Nonetheless, the characteristic decay constant $n^*$ remains above 2000 and corresponds to effective distances beyond 300 $\mu m$ (taking a 140 nm quantum dot spacing). The fast decay for the triangular shuttling is likely originating from the diagonal shuttling step. The tunnel coupling between $QD_2$ and $QD_4$ is low and more challenging to control, due to the absence of a dedicated barrier gate. The low tunnel coupling demands slower ramp times ($t_{ramp} \simeq 36$ ns) for the hole transfer. This increases the dephasing experienced by the qubit during each shuttle and also the time spent close to the (1100)-(1001) charge degeneracy point, where fast spin randomization will likely occur.

Remarkably, we find that the performance achieved for the coherent corner shuttling (as shown in Fig. 4d) are comparable to those of coherent shuttling between neighboring quantum dots. This stems from the performance being limited by dephasing. However, the performance for the CPMG sequence appears inferior when compared to the single echo-pulse sequence. Since the shuttling sequence becomes more complex, we speculate that it is harder to exactly compensate for the change in quantization axes. Imperfect compensation may introduce errors, which are not fully decoupled using the CPMG sequence. Alternatively, simulations shown in Supplementary Fig. 14 suggest that the decoupling achieved using a CPMG sequence depends on the idle time in the initial quantum dots. For an idle time corresponding to a $(2k + 1)\pi$ (with $k$ an integer) phase accumulation, the decoupling achieved using either an ideal echo or a CPMG sequence is very similar. In such a scenario, the effect of imperfect decoupling pulses would become more apparent in a CMPG sequence and would lead to decreased performance.

The performance of the coherent triangular shuttling, displayed in Fig. 4g, fall short compared to the corner shuttling. Yet, the number of shuttles reached remains limited by dephasing as

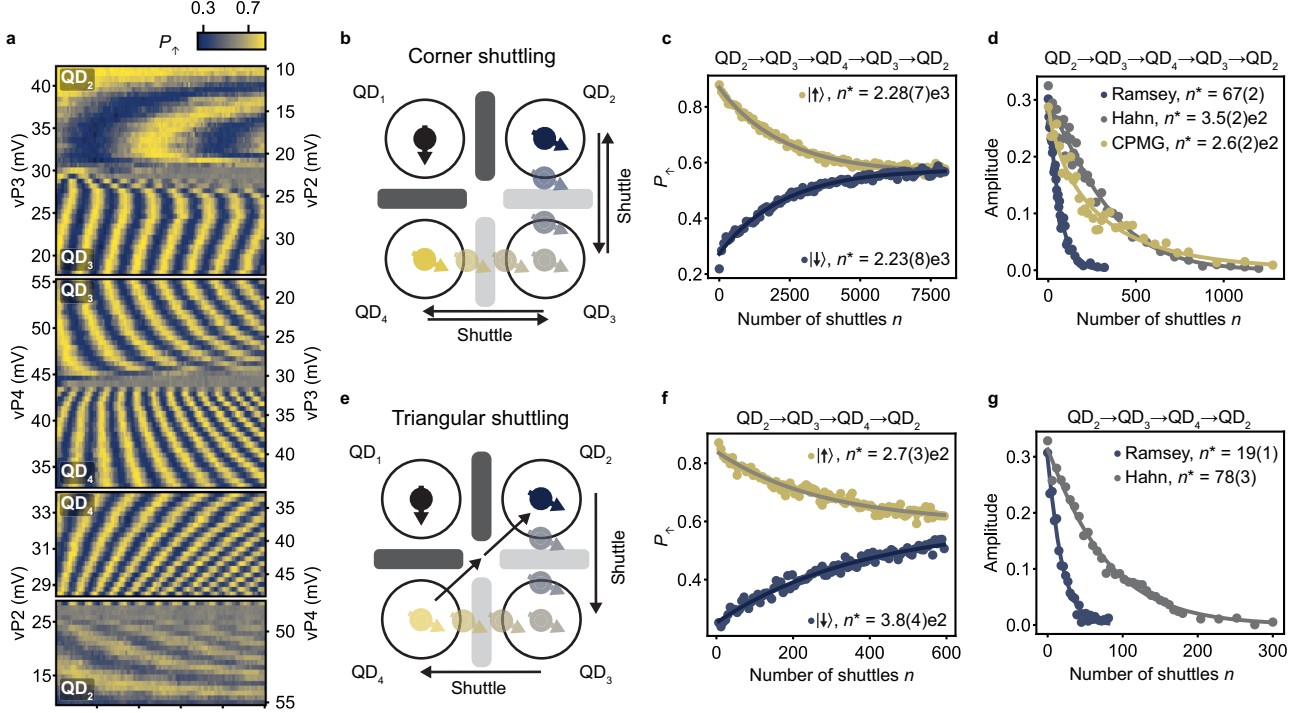

**Fig. 4 | Coherent shuttling through quantum dots. a** Results of free evolution experiments while shuttling through quantum dots (QDs), similar to those displayed in Fig. 1h, I for the corner and triangular shuttling processes. In these experiments, the amplitude of the detuning pulse is increased in steps, in order to shuttle a qubit from $QD_2$ to $QD_3$ and back (top panel), from $QD_2$ to $QD_3$ to $QD_4$ and back (second panel). The measurement in the third panel is identical to the measurement in the second panel, but the final point in the charge stability diagram is stepped towards the charge degeneracy point between $QD_2$ and $QD_4$. In the bottom panel the qubit is shuttled in a triangular fashion: from $QD_2$ to $QD_3$ to $QD_4$ to $QD_2$. The ramp times for this experiment are chosen in such a way that the shuttling is adiabatic with respect to the changes in quantization axis. Schematic illustrating the shuttling of a spin qubit around the corner: from $QD_2$ to $QD_3$ to $QD_4$ and back

via $QD_3$ (**b**) and in a triangular fashion: from $QD_2$ to $QD_3$ to $QD_4$ and directly back to $QD_2$ (**e**). The double arrow from $QD_4$ to $QD_2$ indicates that this pulse is made in two steps, in order for the spin to shuttle via the charge degeneracy point of $QD_4$ - $QD_2$ and avoid crossing charge transition lines. Performance for the corner shuttling (**c**) and the triangular shuttling (**f**) of a qubit prepared in the basis states. The decays of the spin-up probabilities $P_\uparrow$ are fitted (solid lines) by $P_0 \exp(-n/n^*) + P_{sat}$. Performance for shuttling a qubit prepared in a superposition state for the corner shuttling (**d**) and the triangular shuttling (**g**) and for different shuttling sequences. The decays of the coherent amplitude $A$ are fitted (solid lines) by $A_0 \exp(-(n/n^*)^\alpha)$. Shuttling performance for different processes are summarized in Supplementary Table 1. The uncertainties indicate one standard deviation from the best fits.

shown by the large improvement of $n^*$ obtained using dynamical decoupling. The weaker performance are thus predominantly a consequence of the use of longer voltage ramps. A larger number of coherent shuttling steps may be achieved by increasing the diagonal tunnel coupling, which could be obtained by incorporating dedicated barrier gates.

## Discussion

We have demonstrated coherent spin qubit shuttling through quantum dots. While holes in germanium provide challenges due to an anisotropic *g*-tensor, we find that spin basis states can be shuttled $n^* = 2230$ times and coherent states up to $n^* = 67$ times and even up to $n^* = 350$ times when using echo pulses. The small effective mass and high uniformity of strained germanium allow for a comparatively large quantum dot spacing of 140 nm. This results in effective length scales for shuttling basis states of $l_{spin} = 312$ µm and for coherent shuttling of $l_{coh} = 9$ µm. By including echo pulses we can extend the effective length scale to $l_{coh} = 49$ µm. These results compare favorably to effective lengths obtained in silicon[15,27–29]. However, we note that, in general, extrapolating the performance of shuttling experiments over few sites to predict the performance of practical shuttling links requires caution. Quantum dot chains that would allow to couple spin qubits over appreciable length scales will put higher demands on tuning, uniformity, and the ability to tune all the couplings, making the optimization of the shuttling more challenging. Moreover, the spin

dynamics and thus the coherent shuttling performance will depend on the noise in the quantum dot chain. For example, if the noise is local, echo pulses may prove less effective. However, in that case, motional narrowing[22,25,29,45–47] may facilitate the shuttling.

Furthermore, operating at even lower magnetic fields will boost the coherence times[4,37] and thereby increase the shuttling performance. Moreover, at lower magnetic fields the Larmor frequency is lower, which eases the requirements for the precision of the timing of the shuttling pulses. At very low fields, charge noise might not be the limiting noise source anymore and even further improvements may be achieved exploiting purified germanium[4,37,40]. Finally, shuttling could help mitigate problems in qubit addressability which may arise at low magnetic field.

While we have focused on bucket-brigade-mode shuttling, our results also open the path to conveyor-mode shuttling in germanium, where qubits would be coherently displaced in propagating potential wells using shared gate electrodes. This complementary approach holds promise for making scalable mid-range quantum links and has recently been successfully investigated in silicon[29], though on limited length scales. For holes in germanium, the small effective mass and absence of valley degeneracy will be beneficial in conveyor-mode shuttling. Rotations induced by the spin-orbit interaction while shuttling in conveyor-mode could be compensated by applying an appropriate EDSR pulse after the qubit transfer. Such methods could also be used in bucket-brigade-mode shuttling, as suggested by

preliminary experiments shown in Supplementary Note 9. It may allow for even faster qubit transfers and thus shuttling over longer distances.

Importantly, quantum links based on shuttling and spin qubits are realized using the same manufacturing techniques. Their integration in quantum circuits may provide a path toward networked quantum computing.

## Methods

### Materials and device fabrication
The device is fabricated on a strained Ge/SiGe heterostructure grown by chemical vapor deposition[30,48]. From bottom to top the heterostructure is composed of a 1.6-µm thick relaxed Ge layer, a 1-µm step graded $Si_{1-x}Ge_x$ ($x$ going from 1 to 0.8) layer, a 500 nm relaxed $Si_{0.2}Ge_{0.8}$ layer, a strained 16 nm Ge quantum well, a 55 nm $Si_{0.2}Ge_{0.8}$ spacer layer and a < 1-nm thick Si cap. Contacts to the quantum well are made by depositing 30 nm of aluminum on the heterostructure after etching of the oxidized Si cap. The contacts are isolated from the gate electrodes using a 7 nm aluminum oxide layer deposited by atomic layer deposition. The gates are defined by depositing Ti/Pd bilayers. They are separated from each other by 7 nm of aluminum oxide.

### Experimental procedure
To perform the experiments presented, we follow a systematic procedure composed of several steps. We start by preparing the system in a (1,1,1,1) charge state with the hole spins in $QD_1$ and $QD_2$ initialized in a $|\downarrow\rangle$ state, while the other spins are randomly initialized. Subsequently, $QD_3$ and $QD_4$ are depleted to bring the system in a (1,1,0,0) charge configuration. After that, the virtual barrier gate voltage $vB_{12}$ is increased to isolate the ancilla qubit in $QD_1$. The tunnel couplings between $QD_2$ and $QD_3$ and, depending on the experiment, between $QD_3$ and $QD_4$ are then increased by lowering the corresponding barrier gate voltages on $vB_{23}$ and $vB_{34}$. This concludes the system initialization.

Thereafter, the shuttling experiments are performed. In the shuttling experiments, waiting times up to 10 ns are included on both sides of each microwave pulse. These waiting times are short compared to the microwave pulse times as well as the qubit coherence times. Note that to probe the shuttling between $QD_3$ and $QD_4$, the qubit is first transferred adiabatically (with respect to the change in quantization axis) from $QD_2$ to $QD_3$. To determine the final spin state after the shuttling, the qubit is transferred back adiabatically to $QD_2$. Next, the system is brought back in the (1,1,1,1) charge state, the charge regime in which the readout is optimized. This is done by first increasing $vB_{23}$ and $vB_{34}$, then decreasing $vB_{12}$ and finally reloading one hole in both $QD_3$ and $QD_4$. We finally readout the spin state via latched Pauli spin blockade by transferring the qubit in $QD_1$ to $QD_2$ and integrating the signal from the charge sensor for 7 µs. Spin-up probabilities are determined by repeating each experiment a few thousand times. Details about the experimental setup can be found in ref. 2.

### Achieving sub-nanosecond resolution on the voltage pulses
For these experiments, we use voltage pulses applied to the electrostatic gates by the arbitrary wave form generators (AWGs). These pulses are compiled as a sequence of ramps, using a control software. The ramps are defined by high precision floating points: time stamps and voltages. The maximum resolution in time is set by the maximum sample rate of the AWGs, which is 1 GSa/s and which translates to a resolution of 1 ns. Using this sample rate, the signal that is outputted by the AWGs has discrete steps, as depicted in Supplementary Fig. 2a. Simply moving this sampled pulse in time is only possible with a precision of 1 ns. However, it is possible to achieve sub-nanosecond resolution by slightly adjusting the voltages of the pulse instead. As illustrated

in Supplementary Fig. 2a, in this way it is possible to delay a pulse with less than 1 ns. Quantitatively: to achieve a time delay of $\tau$, the voltages forming the ramp are shifted by $-\tau \frac{\mathrm{d}V_{\mathrm{ramp}}(t)}{\mathrm{d}t}$. The output of the AWGs has a higher order low-pass filter with a cut-off frequency of approximately 400 MHz. This filter smoothens the output signal and effectively removes the effect of the time discretization, as is shown in Supplementary Fig. 2b. The time shift of the pulse is not affected by the filter, since it does not change the frequency spectrum of the pulse. To summarize, combining the high precision in the voltages of the pulse with the output filtering of the AWGs allows to output a smooth voltage ramp that is delayed by $\tau < 1$ ns, despite the limited sampling rate. Applying this technique to all voltage ramps results in sub-nanosecond resolution on the overall pulse sequence.

## Data availability
The data generated in this study have been deposited in the Zenodo repository under accession code: https://zenodo.org/records/11203148.

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

## Acknowledgements
We thank A. M. J. Zwerver, M. de Smet, L. M. K. Vandersypen, V. V. Dobrovitski and all the members of the Veldhorst group for inspiring discussions. M.V. acknowledges support through two projectruimtes (No. 680-91-126 and No. 16PR1049) and a Vidi grant (No. 14851), associated with the Netherlands Organization of Scientific Research (NWO), and the ERC Starting Grant QUIST (No. 850641). Research was sponsored by the Army Research Office (ARO) and was accomplished under Grant No. W911NF- 17-1-0274. M.V. also acknowledges financial support from the IGNITE project of European Union's Horizon Europe Framework Programme under grant agreement No. 101069515. The views and conclusions contained in this document are those of the authors and should not be interpreted as representing the official policies, either expressed or implied, of the Army Research Office (ARO), or the U.S. Government. The U.S. Government is authorized to reproduce and distribute reprints for Government purposes notwithstanding any copyright notation herein. This work is part of the 'Quantum Inspire - the Dutch Quantum Computer in the Cloud' project (with project number [NWA.1292.19.194]) of the NWA research program 'Research on Routes by Consortia (ORC)', which is funded by the Netherlands Organization for Scientific Research (NWO). M.R.-R. acknowledges support from the Netherlands Organization of Scientific Research (NWO) under Veni grant (VI.Veni.212.223).

## Author contributions
F.v.R.-D. and C.D. performed the experiments and analyzed the data. C.-A.W. contributed to the experiments, simulations and theoretical analysis. S.L.d.S. wrote the measurement software. W.I.L.L. fabricated the device. M.R.-R. performed the simulations and contributed to the theoretical analysis. N.W.H. contributed to the design and the development of the device as well as the development of the measurement setup. A.S. and G.S. supplied the heterostructures. F.v.R.-D., C.D. and M.V. wrote the manuscript with input from all other authors. M.V. supervised the project.

## Competing interests
N.W.H. declares equity interest in Groove Quantum BV. The other authors declare no competing interests.
