## [Peer Review File · Nature Communications]

REVIEWER COMMENTS

Reviewer #1 (Remarks to the Author):

The manuscript by Floor van Riggelen-Doelman et al. reports spin-qubit shuttling in a 2×2 Ge quantum dot array. Unlike some recent works performed in Si dots, the authors explored non-adiabatic, tailored shuttling pulses to mitigate the adverse effects from strong spin-orbit interactions in Ge dots. Otherwise, the methodologies are largely identical to what was pioneered in Ref. 27 for the demonstration of high-fidelity spin-qubit shuttling in quantum dot systems. However, the technical details of their shuttling performance analysis and the conclusions drawn on the usefulness of diabatic transfer raise questions.

1. I am concerned with the applicability of the non-adiabatic shuttling approach in the future. The authors admit that it is more demanding in the section titled SHUTTLING PERFORMANCE. I suggest that the authors discuss, somewhere in the main text, the implication of having to use sub-nanosecond pulses optimized for individual sites for making quantum links (both in the bucket brigade and conveyer modes).
2. In the non-adiabatic shuttling process between QDs A and B, the basis state in QD A will be transferred to a superposition state in QD B. I am puzzled how the authors can nonetheless separately quantify the shuttling performance for the basis states and superpositions. To give a concrete example, why is it that the spin-up probability in Fig. 3c does not decay (at least initially) due to dephasing in QD3 (during idle time 2)? Put differently, can't we regard the pulse sequence as $(n - 2)$ times QD3 \leftrightarrow QD2 shuttling for a (weighted) superposition state in QD3 with a starting and an ending ramp (from and to QD2)? There must be something that I am missing.
3. In the non-adiabatic regime, the qubit state will be rotated every time it is transferred even though the authors chose the idle time carefully so that it will rotate back (in the absence of noise). Doesn't this refocus shuttling errors partially, as the qubit passes through the same site multiple times during the performance test? (If so, the value n^* should not be compared to the results in earlier works.) I request that the authors explain in detail why the effect of rotations does not lead to underestimation of the errors and whether it may have something to do with the non-unity exponents reported in Fig. 3. (A figure corresponding to Supplementary Fig. 8b may be illustrative for readers.)
4. Unfortunately, I am not convinced by the authors conclusion that coherent shuttling performance is limited by dephasing during free evolution. If I understand correctly, the authors (implicitly)

assume that decoupling pulses do not at all cancel errors due to shuttling itself, but how is this assumption warranted? For that matter, is the result shown in Supplementary Fig. 7 consistent with the conclusion, where the overall decay rates Γ do not seem to change much when the free evolution time in QD3 is more than doubled (e.g. 2.0 ns)?

5. The authors conclude that the spin polarization transfer fidelities depend on the spin state (see e.g. Supplementary Table 1.). What justifies modelling the decay by a single exponential in such situations, rather than using a Markov chain model (as in Ref. 27)? In addition, can the authors give the definition of the error bar?

6. The authors report a large change in the quantization axis between dots. The angle θ between the quantization axes is a highly relevant quantity: it characterizes the diabaticity for a given ramp rate and the spin rotation axis in the transfer process (the latter is also important in the context of other qubit operation modes such as the flopping mode). I naively expect that this angle θ can be most reliably and straightforwardly obtained by performing the state tomography (with the projection pulse applied in the quantum dot towards which the qubit is shuttled). The authors however seek for indirect evidence from the oscillation visibility and the four-level modelling results. Given the relevance, more direct measurements are preferred (especially for QDs 3-4). Can the authors clarify the motivation for their choice?

7. In the current device, the tunnel coupling is ~ 10 GHz. I believe it is insightful to quantify the expected size of the tunnel coupling required to realize adiabatic transfer on the timescale of a nanosecond.

8. Can the authors briefly comment on the validity of using CPMG echo amplitudes to assess the (enhancement of) shuttling fidelity when the channel is not pure dephasing?

9. The authors conclude "The fast decay for the triangular shuttling is likely originating from the diagonal shuttling step." Did the authors try measuring the QD2-QD4 shuttle fidelity to directly conclude on this point?

10. In the Methods section, the authors explain how they achieved sub-nanosecond resolution. However, unfortunately, I could not understand what they mean by "Shifting a ramp by τ results in a shift of the voltages by $-\tau dV(t)/dt$ ". Further illustrations will be of help.

11. Please specify the ramp time used in experiments presented in Fig. 1 early in the main text or in the figure caption.

12. The following typos should be fixed: In the section titled SHUTTLING PERFORMANCE: "tenths of nanoseconds" should read "tens of nanoseconds". In Conclusion, "may proof less effective" should probably read "may prove less effective". In Methods, "by first increasing vB34 and vB34" should be changed to something else (vB23 and vB34?). Also, please double check the values of n in the labels in Supplementary Fig. 8 a - I believe n = 2 should read n = 0 (and n = 4 should be n = 2 etc).

To conclude, while this work certainly deserves publication somewhere, I would not argue for publication in Nature Communications based on my current impression of what it adds to the literature.

Reviewer #2 (Remarks to the Author):

The manuscript by van Riggelen-Doelman demonstrates quantum links in semiconductor spin qubits using single hole spins in the Ge/SiGe 2x2 array of [2]. Single-qubit logic is demonstrated by bucket-brigade shuttling across up to three quantum dots leveraging spin orbit interaction. The fourth quantum dot is used for latched readout. The shuttling is performed also diagonally in 2x2 array even in the absence of a dedicated gate barrier. Given the number of shuttles and the distance between the dots, an effective shuttling length of 312 μm is reached for spin basis states, and 9 μm for coherent shuttling. The latter has been extended to 49 μm by echo pulses.

The manuscript is detailed and well written, the analysis of the measurements is sound, and the described experiments constitute a step into a promising direction and potentially more scalable quantum-dot spin qubit architectures. I would therefore recommend publication, subject to answering a few questions:

- 1) From the caption of Suppl. Figure 1, Why is the ramping partially adiabatic (~ 4 ns ramp) In Fig.1(l) instead of adiabatic (~ 40 ns ramp) as in Fig.1(h)?
- 2) What is causing the faded fringes in Fig.1(g) when the resonant frequency is probed in QD3?
- 3) In Suppl. Figure 1 (a) around $V_{p2} = -15$ mV there are missing fits. Why?

- 4) In Suppl. Table 1, CPMG should be 1.30 instead of 1.3 for consistency with the number of significant digits used in the uncertainty.
- 5) Why are the probability oscillations attributed to the flopping mode operation appear only in Q2, as described in Suppl. Note 1?
- 6) Suppl. Figure 4 caption, c) should be replaced by d)
- 7) Is there an applied magnetic field in Fig.1?
- 8) In the text of Fig.4, one given reason for the CPMG sequence performing worse than echo is the high-frequency noise at the interdot transition. Then why does CPMG perform better in the previous Fig.3(g) and (h)?
- 9) Additionally, the fitting parameter α , its physical interpretation in context of the experiment, and its differences for different pulse sequences could be explained better, if not referenced theoretically.
- 10) Why is Fig.4(a) different from Fig.1(h)? Different tuning?
- 11) In the conclusions, the use of “However” in “However for holes in germanium...” is unclear to me.
- 12) Also in the conclusions section, could the authors clarify the sentence “Furthermore, operating at even lower magnetic fields and exploiting purified germanium will boost the coherence time...” or cite some theoretical or previous experimental work supporting this assertion? Additionally, will going to lower fields reduce individual addressability of the qubits?

Reviewer #3 (Remarks to the Author):

Reviewer #4 (Remarks to the Author):

Nat. Comm. 23-40208-T Review Notes

Spin qubits have made a lot of progress in the past few years with regards to high fidelity operation and scaling up device sizes. They are still, however, just at the front end of figuring out the scaling problems. In addition to designing protocols and devices for operating many-qubit registers, there is a potential need for moving quantum information between registers. In the case of spin-qubits, this may be accomplished by physically moving spins between quantum dots.

This publication constitutes a timely contribution to the field, demonstrating the controlled transfer of hole spin across three sites. Significantly, it establishes the feasibility of achieving this process within the few-nanosecond timescale, while maintaining relatively low error rates. The investigation encompasses a comprehensive analysis of coherence properties during the transfer process, as well as an exploration of the dynamic behavior induced by the pronounced spin-orbit coupling. Furthermore, the study illuminates the critical influence of local electrostatic potentials in germanium on these phenomena.

I recommend the paper for publication, but first have some questions and comments that could help guide revisions to make the manuscript stronger before it is accepted.

Questions/Comments

1. The experiments here all involve round-trip shuttling of the qubit, such that it ends up in the same place that it started. This allows techniques like spin-echo and the 2π precession about the tilted axis in each dot to work. Since the goal is ultimately long-range displacement of the qubit, the authors should comment on how the effects that they have observed and measured here will impact this type of shuttle sequence and what techniques will likely need to be used.
2. The figures are all well put together and easy to understand. One possible suggestion is to scale the colour-bar of the oscillations in Fig. 2 to the Fig. 1 data (or other full visibility oscillations) to help make the point about partial oscillations around the tilted axis. Why does Fig. 2d seem to have even lower visibility (going by the colour-bar range) than Figs 2b,c? Is it just averaged more and therefore has less speckle?
3. The authors should include the coherent superposition shuttling fidelity [$\exp(-1/67)$] alongside the projection fidelity. The echoed fidelity is less useful since non-round-trip shuttling won't really get to take advantage of an echo, but it could be stated as well.

a. It would be good to compare the coherence time during shuttling to the coherence times measured in the individual dots (which should be stated in text or supp somewhere). For instance, if 67 corner-shuttle sequences takes 350 ns is this close to the average $T2^*$ measured in the array?

b. Similarly, it would be nice to see $T1$'s reported and compared with shuttling times for the projections. This seems to not be limited by $T1$ given the decay towards 50% instead of 0%. I expect $T1$ is ~ 1 ms while the shuttle decay is on a timescale that seems closer to 100 microseconds.

4. What causes the spin projection measurements to decay to 50% instead of 0%? Is it from small diabatic transition probabilities, slight errors from the tilted axis problem accumulating, other?

5. Potentially as part of the supplementary discussion of off axis rotations and tunnel coupling estimates: What are the estimated level velocities at the interdot charge transitions? Can one use this to estimate bounds on the tunnel coupling between dots 2 and 4 since that seems to be limited by adiabatic requirements?

a. It would be good to have some discussion, perhaps in supplement, on any interplay between $T2^*$ and tunnel couplings if it is known (do highly tunnel-coupled dots have lower $T2^*$ since the confinement would be lower).

6. The conclusions suggest that operating at lower field and exploiting purified germanium will be the top strategies for boosting coherence times. I'm not wanting to expand the conclusions to discuss these points, but I want to understand why these are the two best options. The authors could respond to these two questions:

a. What sets the lower bound for the magnetic field in these experiments and how much better would coherence times be if the field was reduced? Does this start to harm readout and single-qubit controls?

b. It seems from previous Ge work that charge noise and large spin-orbit coupling/g-factor anisotropy dominate decoherence processes. My understanding is that there is weaker hyperfine coupling in Ge holes. It isn't clear that isotopic purification will provide the same kind of boost to coherence that Si e- qubits received. Is there existing literature that shows coherence times improving in 70Ge heterostructures?

7. Not necessary to have in the manuscript, but could be a good discussion point: For spin shuttling to be a viable part of a processor architecture that moves information between nodes during a computation cycle, the shuttle step will likely be required to meet fault tolerant requirements of $\sim 99\%$ as an "identity" gate. From the measurements performed here, are the authors able to estimate what would be needed to move a spin ~ 10 microns with 99% fidelity given the current coherence times in Ge? Are the waveform generation capabilities and achievable tunnel couplings in typical devices within spec for this or do coherence times in Ge need to get better to have a chance?

8. Does the tilted quantization axis phenomenon investigated here play a role in reducing PSB visibility or does one ramp into readout position slowly enough that this is assumed to be negligible?

a. Along these lines, do they have a good understanding of what limits their measurement visibility? Is it mostly relaxation during readout from spin-orbit effects, measurement circuit limitations like SNR/bandwidth, or both?

Small Revisions

1. Shuttling Performance, paragraph 1: Based on the data shown in Fig 2C, they should change "...ramps of tenths of nanoseconds..." to "...ramps of tens of nanoseconds..."
2. Conclusions, paragraph 1: "...echo pulses may prove less effective."
3. In "Shuttling Performance", the polarization transfer fidelity on corner shuttling is quoted as 99.97% and later in "Shuttling through quantum dots" as 99.96%. From the $n^*=3360$, 99.97% is correct. Even at $n^*=3000$, it rounds to 99.97%. Double check that these should be quoted as the same in the text.
4. In methods: Experiment Procedure, they describe the protocol for adjusting barrier voltages during the experiments. There appears to be a typo in the second paragraph where v_{B34} is mentioned twice in a row where I think one of the mentions needs to be v_{B23} .

REVIEWER COMMENTS

Reviewer #1 (Remarks to the Author):

The manuscript by Floor van Riggelen-Doelman et al. reports spin-qubit shuttling in a 2×2 Ge quantum dot array. Unlike some recent works performed in Si dots, the authors explored non-adiabatic, tailored shuttling pulses to mitigate the adverse effects from strong spin-orbit interactions in Ge dots. Otherwise, the methodologies are largely identical to what was pioneered in Ref. 27 for the demonstration of high-fidelity spin-qubit shuttling in quantum dot systems. However, the technical details of their shuttling performance analysis and the conclusions drawn on the usefulness of diabatic transfer raise questions.

We thank the reviewer for the critical comments which have helped to clarify the manuscript and highlight the impact of this work. Below we provide a point-by-point response addressing all questions. Additionally, we would like to point out that this work represents the first demonstration of spin qubit shuttling through a quantum dot, which to the best of our knowledge has not been demonstrated in Ref. 27 or in any other previous work. Our work also contributes to the understanding of the impact of spin-orbit interactions on shuttling, it is the first realization of spin shuttling with holes, and it is an advancement of the state-of-the-art in quantum information transfer using spin qubit shuttling.

1. I am concerned with the applicability of the non-adiabatic shuttling approach in the future. The authors admit that it is more demanding in the section titled SHUTTLING PERFORMANCE. I suggest that the authors discuss, somewhere in the main text, the implication of having to use sub-nanosecond pulses optimized for individual sites for making quantum links (both in the bucket brigade and conveyer modes).

We thank the reviewer for this question. We would like to point out that spin qubit shuttling is a compelling approach for quantum links, but it can also provide various applications regarding displacing spin qubits over a short range. For example, it may be exploited to increase connectivity or reduce crosstalk. The reviewer is correct that in this work we make use of sub-nanosecond resolution to enable fast shuttling. This approach may be compatible with large-scale systems, but other complementary solutions can be envisioned.

First, we would like to note that CMOS (and cryo-CMOS) electronics is very suited for providing such high time resolution. Since such electronics is considered a key element in large-scale systems, we envision ample room for their integration.

Second, an alternative strategy can be to implement a final single qubit operation using EDSR at the end of the sequence to correct for any rotation during the diabatic spin shuttling. This has also the benefit of dynamical decoupling as discussed in Ref. A. We have performed a new proof-of-principle experiment, the results of which are presented and discussed in Supplementary Note 9. This strategy would also be suitable to correct rotations induced by conveyor-mode shuttling.

Another mitigation strategy can be to operate at lower magnetic field, such that the Larmor frequency is reduced. This would give the additional benefit of increased coherence.

Finally, we believe optimization of the pulses may be automated in the future for shuttling over few sites inside qubit arrays. Looking at the evolution of the spin probabilities as function of the number of shuttling steps, one can determine the waiting times (see for example Supplementary Figure 11) that cancels the effect of rotations induced by spin-orbit interaction. This may only require additional calibration steps.

We have added a discussion in the discussion part of the main text to address this topic.

Ref A : High-fidelity spin qubit shuttling via large spin-orbit interaction, S. Bosco, J. Zou, D. Loss - arXiv preprint arXiv:2311.15970, 2023

2. In the non-adiabatic shuttling process between QDs A and B, the basis state in QD A will be transferred to a superposition state in QD B. I am puzzled how the authors can nonetheless separately quantify the shuttling performance for the basis states and superpositions. To give a concrete example, why is it that the spin-up probability in Fig. 3c does not decay (at least initially) due to dephasing in QD3 (during idle time 2)? Put differently, can't we regard the pulse sequence as $(n - 2)$ times QD3 \leftrightarrow QD2 shuttling for a (weighted) superposition state in QD3 with a starting and an ending ramp (from and to QD2)? There must be something that I am missing.

We assess the performance of basis states and superposition states by preparing the respective state and monitoring their decay as a function of the number of shuttling steps. Experimentally, we observe a significant difference in their performance and their decay can be fitted with a single exponential curve with a characteristic constant.

Superposition states are typically directly sensitive to changes in the qubit energy. Indeed, when there is a finite change in quantization axis, shuttling between quantum dots makes basis states sensitive to dephasing. However, a limited difference in quantization axis results usually in trajectories that only span a small part of the total qubit space. This limits the impact of dephasing. Dephasing can lead to the exponential decay observed but spin randomization at the charge anticrossing can also explain it. We cannot distinguish the effects of the two with our experiments.

We have added an additional paragraph in the part 'Shuttling performance' of the main text to highlight this point and a new section in the supplementary information (Supplementary Note 8).

3. In the non-adiabatic regime, the qubit state will be rotated every time it is transferred even though the authors chose the idle time carefully so that it will rotate back (in the absence of noise). Doesn't this refocus shuttling errors partially, as the qubit passes through the same site multiple times during the performance test? (If so, the value n^* should not be compared to the results in earlier works.) I request that the authors explain in detail why the effect of rotations does not lead to underestimation of the errors and whether it may have something to do with the non-unity exponents reported in Fig. 3. (A figure corresponding to Supplementary Fig. 8b may be illustrative for readers.)

In our experiments, we optimize the waiting time in the quantum dot to which we shuttle to make a $2\pi n$ rotation. Such a rotation will not lead to refocusing. We also tune the waiting time in the starting quantum dot to optimize the performance. In particular, π rotations are expected to reduce sensitivity to noise.

In our experiments, we do observe non-unity decay exponents. It likely stems from the noise in our device and in particular the low-frequency components. We have added a sentence in the 'Shuttling performance'

part of the main text explaining this. We have measured such decays as well for static qubits, in previous work (see ref 2) and in these experiments (see Supplementary Table 2).

The reviewer is also wondering to which extend numbers can be compared. A large spin-orbit interaction may indeed give rise to improvements in long-range shuttling, as studied recently by Bosco et al. (Ref. A). While we would like to be cautious in extrapolating any results on spin shuttling to shuttling large distances, it is anticipated that holes in germanium will have an intrinsic advantage due to the spin-orbit interaction. Thus, we are of the opinion that it is fair to compare the numbers to previous experiments as the decoupling is intrinsically part of the shuttling process. We have included Ref. A and added a sentence in the Discussion of the main text and a full section in the supplementary (Supp. Note 8) to put forward the possible decoupling effect of shuttling in the presence of changes in quantization axis.

4. Unfortunately, I am not convinced by the authors conclusion that coherent shuttling performance is limited by dephasing during free evolution. If I understand correctly, the authors (implicitly) assume that decoupling pulses do not at all cancel errors due to shuttling itself, but how is this assumption warranted? For that matter, is the result shown in Supplementary Fig. 7 consistent with the conclusion, where the overall decay rates n^* do not seem to change much when the free evolution time in QD3 is more than doubled (e.g. 2.0 ns)?

We thank the reviewer for this question. First, we would like to clarify that by ‘dephasing during the free evolution’ we meant dephasing occurring both during the idle times in the dots and while ramping the detuning. We clarified this in the ‘Shuttling performance’ part of the main text.

We conclude dephasing to be the dominant contribution to the error based on the observed improvement due to decoupling pulses, as well as the reduced performance when performing adiabatic shuttling. To provide further evidence that dephasing is the dominant mechanism, we also included a new Figure, Supp. Fig. 8, where we plot the shuttling as function of the time. We find good agreement between the $T2^$ of static qubits and the decay time in the shuttling experiments.*

5. The authors conclude that the spin polarization transfer fidelities depend on the spin state (see e.g. Supplementary Table 1.). What justifies modelling the decay by a single exponential in such situations, rather than using a Markov chain model (as in Ref. 27)? In addition, can the authors give the definition of the error bar?

We thank the reviewer for this comment. We first note that the decay observed for spin up and spin down states and for the different shuttling processes are overall very similar except for the triangular shuttling process. To investigate the referee’s suggestion, we have performed additional simulations in Supplementary Note 8. Our analysis also includes a Markov chain model. We remark that both a single decay and the Markov chain model lead to indistinguishable results showing a fast and a slow exponential decay of the basis states. Mathematically, this can be explained through the Zassenhaus formula $\exp(xA)\exp(xB)=\exp(x(A+B)+O(x^2))$, where A and B are the error generators describing the (imperfect) idling in the two dots.

The error bars correspond to one standard deviation from the best fits. We clarified it in the main text and the supplementary.

6. The authors report a large change in the quantization axis between dots. The angle theta between the quantization axes is a highly relevant quantity: it characterizes the diabaticity for a given ramp rate and the

spin rotation axis in the transfer process (the latter is also important in the context of other qubit operation modes such as the flopping mode). I naively expect that this angle θ can be most reliably and straightforwardly obtained by performing the state tomography (with the projection pulse applied in the quantum dot towards which the qubit is shuttled). The authors however seek for indirect evidence from the oscillation visibility and the four-level modelling results. Given the relevance, more direct measurements are preferred (especially for QDs 3-4). Can the authors clarify the motivation for their choice?

We agree with the reviewers that other approaches may be used to measure the angle θ , albeit these other approaches may not necessarily be more reliable and straightforward. For example, state tomography will require to readout in a different dot pair for a single shuttle step (which was not calibrated in our experiment) or careful calibration in an experiment where the hole is shuttled back and forth. Moreover, such an experiment may also be sensitive to the level of adiabaticity.

7. In the current device, the tunnel coupling is ~ 10 GHz. I believe it is insightful to quantify the expected size of the tunnel coupling required to realize adiabatic transfer on the timescale of a nanosecond.

The diabaticity of the spin transfer cannot be evaluated directly using Landau-Zener formula. It depends not only on the tunnel couplings but also on the g -tensors in the two dots and on the external magnetic field. Consequently, determining the conditions required to have an adiabatic spin transfer given an arbitrary device configuration would require dedicated simulations that are beyond the scope of this work.

8. Can the authors briefly comment on the validity of using CPMG echo amplitudes to assess the (enhancement of) shuttling fidelity when the channel is not pure dephasing?

Dynamical decoupling using CPMG sequences is an effective strategy to overcome dephasing due to low-frequency noise. In our coherent experiments we observe a significant improvement when applying echo sequences, as expected for dephasing due to the presence of low-frequency noise. Indeed, CPMG sequences cannot correct for any type of error, such as high-frequency noise or amplitude errors.

9. The authors conclude "The fast decay for the triangular shuttling is likely originating from the diagonal shuttling step." Did the authors try measuring the QD2-QD4 shuttle fidelity to directly conclude on this point?

Figure 4A shows the shuttling between each quantum dot. The reduced visibility after the diagonal step already indicates a weaker performance. We conclude that the reduced performance in triangular shuttling is caused by the diagonal shuttling step from the comparison between the corner shuttling and triangular shuttling. This conclusion is also consistent with reduced performance due to a lower tunnel coupling along the diagonal and less control.

10. In the Methods section, the authors explain how they achieved sub-nanosecond resolution. However, unfortunately, I could not understand what they mean by "Shifting a ramp by τ results in a shift of the voltages by $-\tau dV(t)/dt$ ". Further illustrations will be of help.

We clarified the explanation and provided an illustration in Supplementary Figure 2.

11. Please specify the ramp time used in experiments presented in Fig. 1 early in the main text or in the figure caption.

We added the ramp times in caption.

12. The following typos should be fixed: In the section titled SHUTTling PERFORMANCE: "tenths of nanoseconds" should read "tens of nanoseconds". In Conclusion, "may proof less effective" should probably read "may prove less effective". In Methods, "by first increasing vB34 and vB34" should be changed to something else (vB23 and vB34?). Also, please double check the values of n in the labels in Supplementary Fig. 8 a - I believe n = 2 should read n = 0 (and n = 4 should be n = 2 etc).

We fixed these typos and errors.

To conclude, while this work certainly deserves publication somewhere, I would not argue for publication in Nature Communications based on my current impression of what it adds to the literature.

We thank the reviewer for their comments, which have helped to clarify the experiments and to highlight the significance of this work: the demonstration of coherent shuttling of holes, the realization of spin qubit shuttling through a quantum dot, and the experimental observation of the impact of spin-orbit interaction on shuttling.

Reviewer #2 (Remarks to the Author):

The manuscript by van Riggelen-Doelman demonstrates quantum links in semiconductor spin qubits using single hole spins in the Ge/SiGe 2x2 array of [2]. Single-qubit logic is demonstrated by bucket-brigade shuttling across up to three quantum dots leveraging spin orbit interaction. The fourth quantum dot is used for latched readout. The shuttling is performed also diagonally in 2x2 array even in the absence of a dedicated gate barrier. Given the number of shuttles and the distance between the dots, an effective shuttling length of 312 μm is reached for spin basis states, and 9 μm for coherent shuttling. The latter has been extended to 49 μm by echo pulses.

The manuscript is detailed and well written, the analysis of the measurements is sound, and the described experiments constitute a step into a promising direction and potentially more scalable quantum-dot spin qubit architectures. I would therefore recommend publication, subject to answering a few questions:

We thank the reviewer for their comments and positive review. Below we provide a point-by-point response to all comments.

1) From the caption of Suppl. Figure 1, Why is the ramping partially adiabatic (~ 4 ns ramp) In Fig.1(l) instead of adiabatic (~ 40 ns ramp) as in Fig.1(h)?

We agree with the reviewers, but also find that the partial diabaticity has a limited effect here, as the oscillations due to the differences in quantization axes are limited. Their visibility is about 0.1 (Supp. Figure 2) and is small compared to that of the dominant phase oscillations (~ 0.5) as explained in captions of Supp. Fig. 1. Furthermore, the oscillations induced by the change in quantization axes will have a frequency of about 1-1.1 GHz. In Fig 1l, the resolution on the time axis is 1 ns corresponding to a sampling frequency of 1 GHz. Therefore, these shuttling-induced oscillations should appear as a slow modulation of the contrast of the phase oscillations and have therefore a limited impact on the overall figure.

2) What is causing the faded fringes in Fig.1(g) when the resonant frequency is probed in QD3?

We thank the reviewer for pointing that. We would like to refrain from speculation as we do not know their origin.

3) In Suppl. Figure 1 (a) around $V_{p2} = -15$ mV there are missing fits. Why?

We originally did not try to fit the frequency sweeps around $V_{p2} = -15$ mV because the data were showing two peaks overlapping instead of a single peak. The double peak structure originates from the variation of the Rabi frequency along the detuning. We have now fitted these data points also with a single Gaussian, like for the other points, to extract the corresponding average resonance frequencies. We note that there is also one fit missing around $V_{p2} = 30.5$ mV. Here an artifact prevents to fit the data.

4) In Suppl. Table 1, CPMG should be 1.30 instead of 1.3 for consistency with the number of significant digits used in the uncertainty.

We thank the reviewer and we have made the notations of the significant digits consistent throughout the tables.

5) Why are the probability oscillations attributed to the flopping mode operation appear only in Q2, as described in Suppl. Note 1?

We thank the reviewer for the question. This mode may appear for all quantum dots, but its appearance will depend on the efficiency of the driving. For example, quantum dot 3 in Fig 1g is being driven by the same amplitude and gate as used to drive in quantum dot 2. The slow driving combined with dephasing in quantum dot 3 likely results in lower visibility of this effect. We have clarified that this effect is not necessarily exclusive to quantum dot 2 in the manuscript.

6) Suppl. Figure 4 caption, c) should be replaced by d)

We thank the reviewer and have corrected this.

7) Is there an applied magnetic field in Fig.1?

Indeed, we operate the device with in an in-plane magnetic field of 0.25 T unless stated otherwise. We have now specified it clearly in the caption of Figure 1.

8) In the text of Fig.4, one given reason for the CPMG sequence performing worse than echo is the high-frequency noise at the interdot transition. Then why does CPMG perform better in the previous Fig.3(g) and (h)?

In general, the CPMG sequence includes both more shuttling pulses and more EDSR pulses than the Ramsey sequence. Therefore, we expect it to be more sensitive to pulse imperfections in particular errors in the timing of the refocussing pulses, that were not calibrated in the experiments to achieve high-fidelity operation. This may lead to reduced performances of the CPMG.

We performed some additional simulations (see Supplementary Note 8) to understand this discrepancy. We discovered that the improvements obtained using the CPMG and echo sequences depend on the waiting time in the initial quantum dot. In the extreme case where we wait for a π time in dot 2, an ideal echo sequence can perform equally good as an ideal CPMG sequence due to the phase flip. This combined with the previous argument about imperfect pulses could explain the discrepancy between the data for the shuttling between the double dots and the corner shuttling.

In the view of these elements, we changed the corresponding part in the section ‘Shuttling through quantum dots’ in the main text.

9) Additionally, the fitting parameter α , its physical interpretation in context of the experiment, and its differences for different pulse sequences could be explained better, if not referenced theoretically.

We thank the reviewer for the suggestion.

We have added additional explanations about the physical meaning of this parameter in the main text, in the part ‘Shuttling Performance’.

10) Why is Fig.4(a) different from Fig.1(h)? Different tuning?

Indeed, the tuning for these experiments is different. In particular, the barrier v_{B34} was closed for Fig 1h while it was opened for Fig. 4a. The g -factor dependence with the electric field explains the slight difference in the evolution of the frequency of the oscillations.

11) In the conclusions, the use of “However” in “However for holes in germanium...” is unclear to me.

We have clarified this statement to highlight that germanium may have benefits for conveyor mode shuttling. We now write:

“This complementary approach holds promise for making scalable mid-range quantum links and has recently been successfully investigated in silicon [29], though conveyor-mode shuttling has only been established over limited length scales. For holes in germanium, the small effective mass and the absence of valley degeneracy may provide beneficial aspects to achieve conveyor-mode shuttling over longer distances.”

12) Also in the conclusions section, could the authors clarify the sentence “Furthermore, operating at even lower magnetic fields and exploiting purified germanium will boost the coherence time...” or cite some theoretical or previous experimental work supporting this assertion? Additionally, will going to lower fields reduce individual addressability of the qubits?

At high magnetic fields, coherence is limited by charge noise that can modulate the g-tensor. This decoherence scales with the magnetic field as shown in refs 37 and 40. At low magnetic field decoherence due to hyperfine interaction will dominate. Purified germanium may then allow for even longer coherence. At such low magnetic fields it is indeed likely that the differences in qubit resonance frequency will be small. We envision that shuttling may facilitate qubit addressability, as qubits can be displaced and separated to avoid cross talk. We have updated our manuscript to improve readability.

Reviewer #3 (Remarks to the Author):

We thank the reviewer for their effort.

Reviewer #4 (Remarks to the Author):

Nat. Comm. 23-40208-T Review Notes

Spin qubits have made a lot of progress in the past few years with regards to high fidelity operation and scaling up device sizes. They are still, however, just at the front end of figuring out the scaling problems. In addition to designing protocols and devices for operating many-qubit registers, there is a potential need for moving quantum information between registers. In the case of spin-qubits, this may be accomplished by physically moving spins between quantum dots.

This publication constitutes a timely contribution to the field, demonstrating the controlled transfer of hole spin across three sites. Significantly, it establishes the feasibility of achieving this process within the few-nanosecond timescale, while maintaining relatively low error rates. The investigation encompasses a

comprehensive analysis of coherence properties during the transfer process, as well as an exploration of the dynamic behavior induced by the pronounced spin-orbit coupling. Furthermore, the study illuminates the critical influence of local electrostatic potentials in germanium on these phenomena. I recommend the paper for publication, but first have some questions and comments that could help guide revisions to make the manuscript stronger before it is accepted.

We thank the reviewer for the comments and recommendation for publication. We agree with the reviewer that spin qubit shuttling may find applications as long-range quantum link. In addition, shorter range spin qubit shuttling may also enable higher connectivity and qubit addressability, providing relevance to a broad study on this subject including the impact of dynamical decoupling. Below we provide a point-by-point response addressing all the comments.

Questions/Comments

1. The experiments here all involve round-trip shuttling of the qubit, such that it ends up in the same place that it started. This allows techniques like spin-echo and the 2π precession about the tilted axis in each dot to work. Since the goal is ultimately long-range displacement of the qubit, the authors should comment on how the effects that they have observed and measured here will impact this type of shuttle sequence and what techniques will likely need to be used.

We agree with the reviewer that long-range quantum links is an appealing application for spin qubit shuttling. Simultaneously, we also envision that short-range spin qubit shuttling in dense arrays with sparse occupation may facilitate an increased connectivity and give more flexibility in device operations. This may provide methods to implement efficient error correction codes or quantum algorithms. We give further examples in the main text that motivate the use of short-range shuttling in future spin qubit devices.

In general, we can envision several strategies that could be followed to implement spin qubit shuttling in such architectures. A first method could be to operate adiabatically, but this may compromise the speed and performance. In our work we mitigated undesired rotations by carefully timing each shuttle step. We believe that this method is suitable for shuttling through few dots. In this case, the calibration of the adapted waiting times could be automated in the future. An alternative method could be to shuttle through the array and only apply a correction gate at the final step. In the Supplementary Note 9, we have included a figure as a proof-of-principle of the concept.

The impact of refocusing pulses will depend on the noise spectrum and type of quantum information that needs to be displaced. However, the presence of strong spin-orbit interaction, may lead to an effective decoupling to noise when shuttling in longer arrays, as is recently predicted by Bosco et al (ref A added in the main text).

We have added a small section on this discussion in the discussion section of the main text and included the reference to high-fidelity shuttling in the presence of strong spin-orbit interaction.

Ref A : High-fidelity spin qubit shuttling via large spin-orbit interaction, S. Bosco, J. Zou, D. Loss - arXiv preprint arXiv:2311.15970, 2023

2. The figures are all well put together and easy to understand. One possible suggestion is to scale the colour-bar of the oscillations in Fig. 2 to the Fig. 1 data (or other full visibility oscillations) to help make the

point about partial oscillations around the tilted axis. Why does Fig. 2d seem to have even lower visibility (going by the colour-bar range) than Figs 2b,c? Is it just averaged more and therefore has less speckle?

We thank the reviewer for the suggestion. However, a direct comparison between the amplitudes is not possible due to difference in the readout performance. This is also the main reason explaining the different visibility observed in Fig. 2d. However, we have added a sentence to the caption to Figure 2 to highlight it.

3. The authors should include the coherent superposition shuttling fidelity [$\exp(-1/67)$] alongside the projection fidelity. The echoed fidelity is less useful since non-round-trip shuttling won't really get to take advantage of an echo, but it could be stated as well.

We thank the reviewer for this suggestion, but as the decay for the shuttling of superposition states (both with and without echo pulses) is not purely exponential, we cannot uniquely define a fidelity per shuttle. We therefore choose to abstain from quoting fidelities and refer to the characteristic decay constants.

a. It would be good to compare the coherence time during shuttling to the coherence times measured in the individual dots (which should be stated in text or supp somewhere). For instance, if 67 corner-shuttle sequences takes 350 ns is this close to the average T_2^* measured in the array?

We thank the reviewer for the suggestion and have added a section answering this question in the Supplementary Material (Supp. Note 3). Indeed, once plotting the evolution of the coherence as function of time instead of the number of shuttles, we remark that the characteristic decay time is comparable to the T_2^ measured for static qubits. We note that there are differences in integration time for the two sets of experiments, which may be relevant for the comparison.*

b. Similarly, it would be nice to see T_1 's reported and compared with shuttling times for the projections. This seems to not be limited by T_1 given the decay towards 50% instead of 0%. I expect T_1 is ~ 1 ms while the shuttle decay is on a timescale that seems closer to 100 microseconds.

The decay observed for shuttling of basis state is indeed not due to relaxation but to a randomization of the spin states. T_1 times measured previously on this device (ref 2) are significantly longer than the decay observed of ~ 8000 shuttles corresponding to $\sim 40 \mu\text{s}$. Also, as mentioned by the reviewer, if relaxation was limiting the performances of the shuttling of basis state, spin-up probabilities would decay to 0.

4. What causes the spin projection measurements to decay to 50% instead of 0%? Is it from small diabatic transition probabilities, slight errors from the tilted axis problem accumulating, other?

We thank the reviewer for this insightful question. To provide further understanding, we have performed additional simulations of the shuttling in the new Supplementary Note 8. We show that the decay of the basis states can be explained by dephasing via high-frequency noise along the tilted quantization axis in dot 3 together with spin randomization processes. The dephasing induced decay will (independent of quantization axis difference for θ not equal to 0° and 180°) approach 1/2. On the other hand, the randomization process will decay towards a mixed state. In Supplementary Figure 3, we show the randomization occurs rapidly at the anticrossing where it is dominated by Johnson noise from the control signals (ref 44). Both phenomena leads to an exponential decay towards 0.5 but we cannot determine their relative contribution.

5. Potentially as part of the supplementary discussion of off axis rotations and tunnel coupling estimates: What are the estimated level velocities at the interdot charge transitions? Can one use this to estimate

bounds on the tunnel coupling between dots 2 and 4 since that seems to be limited by adiabatic requirements?

We thank the referee for this suggestion. We added in the corresponding part, the probability of having a Landau-Zener excitation to the first excited charge state while shuttling between neighboring dots. Overall, they are below 2×10^{-4} (see Supp. Section 2) confirming that the shuttling processes are adiabatic with respect to the charge degree of freedom.

We would refrain from using Landau-Zener probabilities to estimate the tunnel coupling between dot 2 and 4. Indeed, we optimized the ramp times by looking at the shuttling of qubits in a spin down state and searching for the longest decay. This decay is a result of different processes like dephasing induced by low and high-frequency noise and spin randomization occurring nearby the charge transition. We cannot disentangle the different effects and thus we cannot evaluate the contribution of diabatic charge transfers to the decays observed. This prevents us to estimate the tunnel coupling between these two dots.

a. It would be good to have some discussion, perhaps in supplement, on any interplay between T_2^* and tunnel couplings if it is known (do highly tunnel-coupled dots have lower T_2^* since the confinement would be lower).

We thank the reviewer for the question. Although we did not perform an extensive study to investigate this effect, we agree that it is to be expected that when the tunnel coupling increases, the T_2^ decreases, because the spin can couple more strongly to its environment. However, since for hole spins in germanium the g -tensor already depends strongly on the electric field, all changes in the gate voltages are likely to have an effect on the coherence of the spin. How strong this effect is, scales with the change of the Larmor frequency with respect to the gate voltage (ref 37). This is also the case for the barrier gate between the quantum dots and it would be non-trivial to disentangle this effect from the effect of the tunnel coupling on the T_2^* .*

In the Supplementary we now added Supplementary Figure 8 which shows the T_2^ for the individual dots, measured at the voltage settings at which the different shuttling experiments were performed. Here we see that the T_2^* of QD3 is higher for the voltage settings of the shuttling experiment QD3-QD4 than for the settings of the other shuttling experiments. One of the differences was the voltage applied to the barrier gate v_{B23} , which was less negative for the shuttling experiment QD3-QD4. This is in line with the idea that if the tunnel coupling is lower that the T_2^* is longer. However, since other voltage settings were also slightly different and because of the before mentioned argument about the g -tensor, it is not possible to relate this difference directly to a difference in tunnel coupling.*

6. The conclusions suggest that operating at lower field and exploiting purified germanium will be the top strategies for boosting coherence times. I'm not wanting to expand the conclusions to discuss these points, but I want to understand why these are the two best options. The authors could respond to these two questions:

a. What sets the lower bound for the magnetic field in these experiments and how much better would coherence times be if the field was reduced? Does this start to harm readout and single-qubit controls?

We worked here at a magnetic field of 0.25 T. This value was chosen because it gives a resonance frequency of about 1 GHz. With the configuration of our hardware at that time, it was the lowest frequency at which we could drive our qubits effectively.

At high magnetic fields, coherence is limited by charge noise that can modulate the g-tensor. This decoherence scales with the magnetic field (ref 37 and 40). At low magnetic field, decoherence due to hyperfine interaction will dominate. Purified germanium may then allow for even longer coherence. We have added a sentence in the Discussion of the main text to clarify this. At such low magnetic fields, it is indeed likely that the differences in qubit resonance frequency will be small. We envision that shuttling may facilitate qubit addressability, as qubits can be displaced and separated to avoid cross talk. We also expect the readout fidelity to improve at lower magnetic field, as relaxation at the anticrossing is reduced.

b. It seems from previous Ge work that charge noise and large spin-orbit coupling/g-factor anisotropy dominate decoherence processes. My understanding is that there is weaker hyperfine coupling in Ge holes. It isn't clear that isotopic purification will provide the same kind of boost to coherence that Si e-qubits received. Is there existing literature that shows coherence times improving in 70Ge heterostructures?

The reviewer is correct that in our experiments at high magnetic field, decoherence is charge-noise limited. However, by operating at low magnetic field, a transition is expected where hyperfine interaction will dominate (see refs 37 and 40). It is currently an open question how long the coherence of purified germanium holes can be at very low magnetic fields.

7. Not necessary to have in the manuscript, but could be a good discussion point: For spin shuttling to be a viable part of a processor architecture that moves information between nodes during a computation cycle, the shuttle step will likely be required to meet fault tolerant requirements of ~99% as an "identity" gate. From the measurements performed here, are the authors able to estimate what would be needed to move a spin ~10 microns with 99% fidelity given the current coherence times in Ge? Are the waveform generation capabilities and achievable tunnel couplings in typical devices within spec for this or do coherence times in Ge need to get better to have a chance?

While we envision multiple use cases for spin qubit shuttling in architectures, achieving a high-fidelity link between remote qubits separated more than 10 microns would be an important milestone. In our work we demonstrate for the first time that a spin qubit can be shuttled through a quantum dot, but we do not meet the performance required for this milestone. However, we believe such a milestone can be achieved by mitigating the effects of dephasing. We envision that this can be done implementing two complementary strategies. The first is to operate at lower magnetic fields (possibly in combination with purified germanium) to increase the coherence. Secondly, for this application, we envision it is important to shuttle fast to combat decoherence. Rather than waiting for a 2π rotation, we envision that a direct shuttling together with a final correction pulse will allow to substantially increase the distance over which coherently can be shuttled. We have included in the Supplementary Note 9, a figure highlighting the proof-of-principle. Finally, we note that motional narrowing and random decoupling effects induced by the strong spin-orbit interaction (ref A) may help in shuttling coherently over long distances. In the discussion section of the main text we have added several sentences on how to improve the shuttling performance even further.

We are optimistic that this work provides a path to obtain a high-fidelity quantum link between remote qubits.

8. Does the tilted quantization axis phenomenon investigated here play a role in reducing PSB visibility or does one ramp into readout position slowly enough that this is assumed to be negligible?

This is a good question and readout in germanium is challenging. While obtaining PSB is simplified compared to silicon because of the absence of valley states, the anisotropic g-tensor (which gives rise to the tilt in quantization axis) results in significant and unpredictable anticrossings between the spin states (see ref B for example). The optimization of the readout ramp thus is dependent on the g-tensors configuration in each device and on the orientation of the external magnetic field (that we cannot tune in our setup).

*Ref B : Classification and magic magnetic field directions for spin-orbit-coupled double quantum dots, A. Sen , G. Frank, B. Kolok, J. Danon , and A. Pályi, PRB, **108**, 245406 (2023)*

a. Along these lines, do they have a good understanding of what limits their measurement visibility? Is it mostly relaxation during readout from spin-orbit effects, measurement circuit limitations like SNR/bandwidth, or both?

Several factors can limit the measurement fidelity, including variations in the g-tensor and relaxation. In our experiments we make use of RF SHTs which provide a good SNR after integrating $\sim 7\mu\text{s}$. Yet, we still think that our readout/initialization fidelity is limiting the measurement fidelity. While improving the charge readout may help, we envision that the main improvement could come from systems where g-tensors can be engineered at will. This may require device fabrication that minimizes strain variations coming from the gates. Again, working at lower magnetic field may help to increase the readout fidelity.

Small Revisions

1. Shuttling Performance, paragraph 1: Based on the data shown in Fig 2C, they should change "...ramps of tenths of nanoseconds..." to "...ramps of tens of nanoseconds..."
2. Conclusions, paragraph 1: "...echo pulses may prove less effective."

We corrected these typos.

3. In "Shuttling Performance", the polarization transfer fidelity on corner shuttling is quoted as 99.97% and later in "Shuttling through quantum dots" as 99.96%. From the $n^*=3360$, 99.97% is correct. Even at $n^*=3000$, it rounds to 99.97%. Double check that these should be quoted as the same in the text.

We thank the reviewer for this comment. However, we checked and the numbers for the fidelities in the text are correct. In the section "Shuttling Performance" we are discussing the performance of the basis state shuttling in the two double dot pairs. Here the typical number n^ that we can shuttled is on average 3140, which corresponds to a fidelity of 99.97%. In the next section we introduce the shuttling through quantum dots, i.e. the corner shuttling and the triangular shuttling. There the typical number n^* that we find for the corner shuttling (on average, for basis state shuttling) is 2255. This corresponds to a fidelity of 99.96%.*

4. In methods: Experiment Procedure, they describe the protocol for adjusting barrier voltages during the experiments. There appears to be a typo in the second paragraph where vB34 is mentioned twice in a row where I think one of the mentions needs to be vB23.

We corrected this typo.

REVIEWER COMMENTS

Reviewer #1 (Remarks to the Author):

I appreciate the revisions made by the authors in response to the reviewers' comments. This work presents the first demonstration of hole spin qubit shuttling through quantum dots and is the first to reveal and address experimentally the challenges due to strong spin-orbit coupling in the qubit shuttling process. I continue to believe that this work deserves publication after validation of the performance characterization methods, although I do not necessarily argue strongly for publication in Nature Communications. I do not support the claim made by the authors in the response letter that "this work represents the first demonstration of spin qubit shuttling through a quantum dot, which to the best of our knowledge has not been demonstrated in Ref. 27 or in any other previous work". I speculate that it simply contains some typographical errors – but in case it does not, from my perspective, Refs. 27 and 15 have demonstrated spin qubit shuttling through a quantum dot (in a phase coherent manner) and there are many other works in the reference list showing spin shuttling through quantum dots in a broader sense (without necessarily preserving quantum information).

With that being said, the authors have addressed most of the issues raised in the previous peer-review round. In my opinion, the following points remain to be further clarified before publication. In particular, I am not yet convinced of the validity of the claimed value of transfer fidelity for the basis state as a performance metric and wonder the impact of the idle time in the starting dot for performance characterization (both basis- and superposition-state shuttling).

1. While I appreciate the new section in the supplementary information (Supplementary Section 8), I continue to be puzzled how the authors claim to separately quantify the shuttling performance for the basis states and superpositions in a relevant manner (this is a follow up on comment #2 in the previous round). In the present work, spin states are non-adiabatically shuttled between dots with respect to the difference in the quantization axis. Thus, a basis spin-state in the starting dot is in general transferred to a non-basis (superposition) state with respect to the quantization axis in the following dots, which exposes the transferred "basis" spin-state to a dephasing channel while in the other dots. This unusual dephasing effect induces the rapid initial decay in the green curves in Supplementary Fig. 13 and may be the cause of initial "outlier" points in Figs. 3 c, d and Figs. c, f. In my current understanding, the depolarization rate obtained in the demonstrated methodology does not give a relevant performance metric of non-adiabatic spin shuttling in practical scenarios. My main concerns are detailed below.

a. In their response, the authors said: "However, a limited difference in quantization axis results usually in trajectories that only span a small part of the total qubit space. This limits the impact of dephasing." While this may suffice when the claimed transfer fidelity is low (with respect to the residual dephasing effect when refocusing may be used and ~75% for the present device parameter

in simulations where refocusing cannot be applied), I do not see why the dephasing effect does not impact the claimed fidelity of ~99.97%.

b. (This is a follow up on comment #3 in the previous round.) In the present experiments, this dephasing effect on the transferred "basis" state is echoed out by shuttling the spin back to the starting dot frequently (with the idle time chosen to yield a finite phase rotation) – this is now explained in Supplementary Note 8B ("we believe that our timing is chosen such that the fast initial Gaussian dephasing is echoed out [10]"). Is this noise cancellation mechanism relevant to shuttling in practice, without the qubit having to pass through the same site multiple times frequently or assuming spatial noise correlations stronger than what is reported? (I would like to understand how the presented characterization method, performed without the idle time in the starting dot chosen to yield net zero phase accumulation (i.e. m is not an even integer), can properly estimate the basis state transfer fidelity without refocusing the low-frequency dephasing error that would be present in realistic scenarios of qubit shuttling within a quantum dot array.)

2. Related to the above and comment #5 in the previous round, what is the message of the two n^* values shown in the figures for the basis-state shuttling characterization results? My understanding is that these numbers should not be considered to represent the individual (state-dependent) relaxation rates, since n^* characterizes the average transfer fidelity and is thus supposed to be identical for (initially) spin-up and -down traces. The state dependence of transfer fidelities should be assessed separately, e.g. for a simple case of a single exponential decay, from the settling value after correction for visibility (using e.g. a Markov chain model). Does the model used by the authors predict the different values of n^* for the spin-up and -down cases?

3. Supplementary Note 8B reveals the significant effect of the starting dot idle time ($\langle \phi_{\text{QD2}} \rangle$ or m) on the basis-state shuttling and the echo and CPMG decoupling. A similar discussion needs to be included for Ramsey shuttling experiments.

4. My understanding is that the qubit phase accumulation in the dots is defined in the lab frame, whereas the X and Y pulses used in the Hahn echo and CPMG experiments are with respect to the qubit (or microwave) frame in the starting dot. I suggest that the authors explicitly state this somewhere in the main manuscript to improve readability.

5. To improve readability, I suggest that the authors explain more explicitly the distinction between adiabaticity with respect to the charge and spin degrees of freedom in the main text.

6. I find it hard to understand the following sentences (before Eq. (12) in the supplementary material): "... and we assume a negligible relaxation in quantum dot QD₂ such that $\gamma_{r,QD2} = 0$. This allows us to allocate the phase accumulations during the adiabatic (diabatic) part of the ramp to $\gamma_{\phi,QD2}$ ($\gamma_{\phi,QD2}$)." What justifies the assumption $\gamma_{r,QD2} = 0$ (near the charge anticrossing)? Do the authors assume $\gamma_{r,QD3} = 0$ as well? What is the role played by $\gamma_{\phi,QD3}$?

7. Do the authors imply that $\theta_{24} = \theta_{34} + \theta_{23}$ in Eq. (18)? If so, why is this the case, since naively σ_y would be required when a third dot is involved?

8. I suggest that the authors double check the following parts for typographical/grammatical errors.

a. on Page 4: "finite time spent in each quantum dots"

b. in Supplementary Fig. 3 a: "instead what visible here is an aliasing pattern"

c. in Supplementary Fig. 3 b: "The error bar correspond to"

d. in Supplementary Note 2B: "We emphasize that the factor 2 in front $\Delta\epsilon_{ij}$ " and "shuttling between QD₂ and QD₃,"

e. in Supplementary Note 8B: "Since, our experimental results"

f. in Supplementary Note 9: "the spin does not experienced"

Reviewer #2 (Remarks to the Author):

The authors have revised the manuscript to a great extent, including changes in the figure, text and supplementary sections. I believe they address my comments and those of the other reviewers satisfactorily.

The impact of high-fidelity shuttling of spin qubits and of quantum information in general is expected to be high, and the paper is of good quality and represents an important step in this area. I therefore would now recommend the paper for publication.

One more typo: Supplementary Note 8 reads "This could possible".

Reviewer #3 (Remarks to the Author):

Reviewer #4 (Remarks to the Author):

The authors addressed the review comments and I am comfortable recommending the manuscript for publication.

REVIEWER COMMENTS

Reviewer #1 (Remarks to the Author):

I appreciate the revisions made by the authors in response to the reviewers' comments. This work presents the first demonstration of hole spin qubit shuttling through quantum dots and is the first to reveal and address experimentally the challenges due to strong spin-orbit coupling in the qubit shuttling process. I continue to believe that this work deserves publication after validation of the performance characterization methods, although I do not necessarily argue strongly for publication in Nature Communications.

We thank the reviewer for the critical appraisal. As we will explain below, we believe the reviewer has missed a critical aspect of our work, and we have further clarified this in the manuscript.

I do not support the claim made by the authors in the response letter that "this work represents the first demonstration of spin qubit shuttling through a quantum dot, which to the best of our knowledge has not been demonstrated in Ref. 27 or in any other previous work". I speculate that it simply contains some typographical errors – but in case it does not, from my perspective, Refs. 27 and 15 have demonstrated spin qubit shuttling through a quantum dot (in a phase coherent manner) and there are many other works in the reference list showing spin shuttling through quantum dots in a broader sense (without necessarily preserving quantum information).

We thank the reviewer for making this comment as this is an important aspect of our work. Within a chain, a quantum dot will have at least two neighbours, to transport a spin state from one to another location. While reference 27 and 15 are important developments, in those works shuttling is studied in a double-quantum dot system only. We referred to this as shuttling between quantum dots, whereas we meant with shuttling through a quantum dot the scenario of shuttling through a quantum dot in a chain. We would like to point out that the difference is more than semantics: it requires the tuning of more parameters, and it is more demanding on the required uniformity. Moreover, more noise coming from different sources can couple to the system, while in a chain there can be a remaining tunnel coupling (e.g. when shuttling from A to B there may be a partial coupling of B to C). All of these may affect the performance and thus establishing shuttling through a quantum dot in a chain is an important advancement.

To make this point more prominent we now write:

Introduction, third paragraph: Networked quantum computers, however, will require integration of qubit control and shuttling through chains of quantum dots, incorporating quantum dots that have at least two neighbours.

Introduction, fifth paragraph: Importantly, we operate in a regime where we can implement single qubit logic and coherently transfer spin qubits through an intermediate quantum dot.

Page 6: For distant qubit coupling, it is essential that a qubit can be coherently shuttled through chains of quantum dots. This is more challenging, as it requires control and optimization of a larger amount of parameters, while more noise sources may couple to the system. Within a chain, a quantum dot will have at least two neighbours, to transport spin states from one site to another passing by intermediates quantum dots. Therefore, an array of three quantum dots could be considered as the minimum size to explore the performance of shuttling in a chain.

With that being said, the authors have addressed most of the issues raised in the previous peer-review round. In my opinion, the following points remain to be further clarified before publication. In particular, I am not yet convinced of the validity of the claimed value of transfer fidelity for the basis state as a performance metric and wonder the impact of the idle time in the starting dot for performance characterization (both basis- and superposition-state shuttling).

The transfer fidelity in our experiment relates to the number of times a state can be shuttled and a higher fidelity indicates that we can shuttle more often. It is an open question to which extend such numbers can

be used to predict the performance in a realistic chain. While in our work we satisfy the condition that at least one quantum dot has two neighbours, the noise present in a longer chain may be different and indeed lead to different results. Rather than speculating we would like to make this point clear and write in the conclusion:

Page7: These results compare favourably to effective lengths obtained in silicon. However, we note that, in general, extrapolating the performance of shuttling experiments over few sites to predict the performance of practical shuttling links requires caution. Quantum dot chains that would allow to couple spin qubits over appreciable length scales will put higher demands on tuning, uniformity, and the ability to tune all couplings, making the optimization of the shuttling more challenging. Moreover, the spin dynamics and thus the coherent shuttling performance will depend on the noise in the quantum dot chain.

Furthermore, we emphasize that the fidelities extracted with our protocol is an averaged quantity. It is questionable whether the number obtained can be extrapolated or interpolated to predict the fidelity of a single shuttle between two quantum dots. It rather quantifies the average error induced by one shuttle inside a sequence incorporating several shuttles. This is intrinsic to any benchmarking technique relying on error amplification and similar discussions could hold, for example, for the interpretation of the gate fidelity extracted from randomized benchmarking. Despite that, we believe that this fidelity is still a relevant quantity to evaluate the performance of shuttling process and to compare our results to previous work that use similar characterization techniques (for example in ref 15 and 27).

To clarify this point, we have changed the word "fidelity" by "*corresponding fidelities per shuttle within the sequence*" in the main text.

1. While I appreciate the new section in the supplementary information (Supplementary Section 8), I continue to be puzzled how the authors claim to separately quantify the shuttling performance for the basis states and superpositions in a relevant manner (this is a follow up on comment #2 in the previous round). In the present work, spin states are non-adiabatically shuttled between dots with respect to the difference in the quantization axis. Thus, a basis spin-state in the starting dot is in general transferred to a non-basis (superposition) state with respect to the quantization axis in the following dots, which exposes the transferred "basis" spin-state to a dephasing channel while in the other dots. This unusual dephasing effect induces the rapid initial decay in the green curves in Supplementary Fig. 13 and may be the cause of initial "outlier" points in Figs. 3 c, d and Figs. c, f. In my current understanding, the depolarization rate obtained in the demonstrated methodology does not give a relevant performance metric of non-adiabatic spin shuttling in practical scenarios. My main concerns are detailed below.

We believe the outlier points mentioned may have a different origin than the fast initial decay highlighted in Supplementary Section 8. Indeed, we also observe such outliers for the polarization decays measured in adiabatic shuttling experiments (Supp. Fig. 9.c). Likewise, in the different plots, we do not observe these outlier points for both decays as one would expect for a fast decay induced by dephasing. The outliers may rather originate from changes in readout visibility induced by the application of the voltage pulses used to shuttle the qubits.

a. In their response, the authors said: "However, a limited difference in quantization axis results usually in trajectories that only span a small part of the total qubit space. This limits the impact of dephasing." While this may suffice when the claimed transfer fidelity is low (with respect to the residual dephasing effect when refocusing may be used and ~75% for the present device parameter in simulations where refocusing cannot be applied), I do not see why the dephasing effect does not impact the claimed fidelity of ~99.97%.

The referee is correct in stating that the dephasing impacts the shuttling of basis states. When optimizing the shuttling performances, we carefully tuned all the idling times and ramp times in the shuttling pulses to achieve the best performance. As explained in Supplementary Note 8B, the idle time in the initial quantum dot may have been chosen such that the initial fast Gaussian dephasing is echoed out. Such a decoupling effect is rather generic as it is likely to occur in a large majority of the choices of the waiting times in the

starting quantum dots as evidenced by new simulations now described in Supplementary Note 8 and plotted in Supplementary Figure 13. The exponential decay that we observe and the high performance that we achieve for shuttling a spin qubit initialized in a basis state suggests that the effect of dephasing was mitigated in our experiments resulting in a high-fidelity.

To highlight this point further, we have added a sentence in the main text:

Page 4: "In our experiment, the dephasing is probably mitigated by a decoupling effect induced by repeatedly waiting in the initial quantum dot (see explanation Supplementary Note 8)."

We have expended the part related to the influence of the choice of the waiting time in the first dot in the Supplementary Note 8. It emphasizes that the decoupling effect always occurs, except for waiting times that correspond to rotation very close to (a multiple of) 2π in the first dot. Therefore, our experiments account for the most generic scenario.

b. (This is a follow up on comment #3 in the previous round.) In the present experiments, this dephasing effect on the transferred "basis" state is echoed out by shuttling the spin back to the starting dot frequently (with the idle time chosen to yield a finite phase rotation) – this is now explained in Supplementary Note 8B ("we believe that our timing is chosen such that the fast initial Gaussian dephasing is echoed out [10]"). Is this noise cancellation mechanism relevant to shuttling in practice, without the qubit having to pass through the same site multiple times frequently or assuming spatial noise correlations stronger than what is reported? (I would like to understand how the presented characterization method, performed without the idle time in the starting dot chosen to yield net zero phase accumulation (i.e. m is not an even integer), can properly estimate the basis state transfer fidelity without refocusing the low-frequency dephasing error that would be present in realistic scenarios of qubit shuttling within a quantum dot array.)

We first note that there can be multiple realistic applications of spin qubit shuttling. Even shuttling between two quantum dots can have a use case to create addressability or e.g. to establish lower crosstalk in an array with sparse occupation. For such applications, the qubits may regularly go back to its original position leading to a decoupling effect.

We do agree with the reviewer that one should be cautious in extrapolating results to long chains as we mention in the manuscript or to estimate the fidelity of a single shuttling step.

As stated by the referee and as suggested by our simulations, we expect our benchmarking technique to be relevant in the presence of spatially correlated noise. Indeed, in this case refocusing effects will also occur while shuttling as suggested in the recent work of ref. 47. The efficiency of these effects will depend on the strength of the noise correlations. Recent experiments of ref. A suggest that noise correlations between neighboring dots can actually be significant.

Alternatively, our method remains fully relevant for adiabatic shuttling with respect to the change in quantization axis. In particular, we expect it to be useful for characterizing shuttling in the presence of magnetic fields with out-of-plane components that would lead to an alignment of the quantization axes.

We have clarified these points by adding the following sentences in the main text:

Page 4: "While extrapolating this result to a long chain of quantum dots is not straightforward, similar noise-averaging effects may occur in the presence of spatially correlated noise in the chain. In the absence of decoupling effects and for the purpose of shuttling basis states, adiabatic shuttling still provides a good alternative as we find n^ to remain above 1000 corresponding to fidelities per shuttle within the sequence above 99.90 % (see Supplementary Figure 9)."*

Ref A: J. Yoneda et al., "Noise-correlation spectrum for a pair of spin qubits in silicon", *Nature Physics* **19**, 1793-1798, 2023

2. Related to the above and comment #5 in the previous round, what is the message of the two n^* values shown in the figures for the basis-state shuttling characterization results? My understanding is that these numbers should not be considered to represent the individual (state-dependent) relaxation rates, since n^* characterizes the average transfer fidelity and is thus supposed to be identical for (initially) spin-up and -down traces. The state dependence of transfer fidelities should be assessed separately, e.g. for a simple case of a single exponential decay, from the settling value after correction for visibility (using e.g. a Markov chain model). Does the model used by the authors predict the different values of n^* for the spin-up and -down cases?

The n^* values should be understood as a simple quantitative description of the measurement results. We report two values of n^* because the data for the shuttling of qubits initialized in the spin-up state and for the shuttling of qubits initialized in the spin-down state were obtained in two separate sets of experiments. We thus decided to analyze them separately as the device performance and the noise may have change slightly between the two experiments, leading to slightly different n^* .

We agree with the reviewer and expect these two n^* values to be the same. By analyzing the data in this way, we can confirm that indeed, the two values are in good agreements (taking into account the error bars) without making any assumption.

We have added a statement to clarify this point:

Page 4: We extract the characteristic decay constants n^ by fitting the data for the shuttling of qubits prepared in $|\uparrow\rangle$ and $|\downarrow\rangle$ separately as they originate from distinct sets of experiments.*

3. Supplementary Note 8B reveals the significant effect of the starting dot idle time ($\langle\phi_{\text{QD2}}\rangle$ or m) on the basis-state shuttling and the echo and CPMG decoupling. A similar discussion needs to be included for Ramsey shuttling experiments.

In Supplementary Note 8, we have added simulations for Ramsey experiments in the corresponding section and study the influence of the phase accumulated by waiting in the first quantum dots. The simulated data is plotted in Supplementary Figure 14. We observe that the waiting time in the initial quantum dot has little impact on the shape of the decay curve except for a phase accumulated close to (a multiple of) 2π . In this case, the amplitude decays to a finite value instead of zero. Most importantly, we observe that the characteristic decay constant remains the same. We remark that the decay to a finite value disappears when taking into account phase fluctuations of the microwave source. This has been checked numerically.

4. My understanding is that the qubit phase accumulation in the dots is defined in the lab frame, whereas the X and Y pulses used in the Hahn echo and CPMG experiments are with respect to the qubit (or microwave) frame in the starting dot. I suggest that the authors explicitly state this somewhere in the main manuscript to improve readability.

Indeed, the echoing pulses are defined with respect to the rotating frame of the initial quantum dot. The phase accumulated by the qubit in these experiments can be equivalently defined either with respect to the rotating frame or the lab frame.

We have clarified these points in the manuscript by adding the following sentences:

Page 4: Importantly, one must account for z-rotations experienced by the qubits during the experiments and the corresponding phase accumulation defined with respect to the qubit rotating frame in the initial quantum dots. Note that the latter can be equivalently defined with respect to the lab frame.

Page 5: We note that the echoing pulses are defined with respect to the rotating frame of the qubit in the starting quantum dots.

5. To improve readability, I suggest that the authors explain more explicitly the distinction between adiabaticity with respect to the charge and spin degrees of freedom in the main text.

We have added the following sentences to clarify this point (the end of first paragraph of “Coherent shuttling of single hole spin qubits”):

Page 2: The hole carrying the spin remains in its orbital ground state and with increasing $|\epsilon|$, the charge becomes localized in the quantum dot with the lowest chemical potential as displayed in Fig. 1.b. In our experiments, we tune to have adiabatic evolution with respect to charge, and study adiabatic and diabatic shuttling with respect to spin.

Page 3-4: Therefore, rapid shuttling of a hole results in a change of angle between the spin state and the local spin quantization axis. In particular, a qubit in a basis state in QD2 becomes a qubit in a superposition state in QD3 when it is shuttled diabatically with respect to the change in quantization axis.

6. I find it hard to understand the following sentences (before Eq. (12) in the supplementary material): "... and we assume a negligible relaxation in quantum dot QD₂ such that $\gamma_{r,QD2} = 0$. This allows us to allocate the phase accumulations during the adiabatic (diabatic) part of the ramp to $\gamma_{\phi,QD2}$ ($\gamma_{\phi,QD2}$).". What justifies the assumption $\gamma_{r,QD2} = 0$ (near the charge anticrossing)? Do the authors assume $\gamma_{r,QD3} = 0$ as well? What is the role played by $\gamma_{\phi,QD3}$?

We thank the reviewer for pointing us to this misunderstanding. In our analysis, we have separated the dynamics of the qubit into three respective regions. Region I (QD2) accounts for the dynamics far away from the anti-crossing in QD2, region III (QD3) accounts for the dynamics far away from the anti-crossing in QD3, and region II (r,QD2/3) considers the dynamics of moving through the anti-crossing. Due to the long T₁ times in region I and III, spin relaxation effects can be safely ignored, and relaxation (as well as Markovian dephasing) only has to be considered in region II. We have changed the notation of the relaxation rate to avoid confusion.

7. Do the authors imply that $\theta_{24} = \theta_{34} + \theta_{23}$ in Eq. (18)? If so, why is this the case, since naively σ_y would be required when a third dot is involved?

The referee is correct as one would expect a σ_y to account for a rotation angle in the x-y plane (from the viewpoint of QD3, i.e, setting the quantization axis of QD3 in z-direction) for shuttling beyond two quantum dots. However, when using the Euler-angle decomposition, the rotation for shuttling from QD3 to QD4 can be decomposed into a rotation around the quantization axis in QD3, a rotation around the y-axis (from the viewpoint of reference frame, QD2, i.e, setting the quantization axis of QD2 in z-direction, and rotates the spin further in the xz-plane), and another rotation around the quantization axis in QD3. Fortunately, a rotation around the quantization axis in QD3 corresponds to idling in QD3 before shuttling in QD4, and therefore can be accounted for by the idling phase. Since our experiment does not allow us to determine the axes, we have assumed in our simulations that all quantization axes lay in the same plane and checked numerically that the impact of this misalignment is negligible in general. We have included a paragraph in the supplement after Eq. (19) discussing this.

8. I suggest that the authors double check the following parts for typographical/grammatical errors.

- a. on Page 4: "finite time spent in each quantum dots"
- b. in Supplementary Fig. 3 a: "instead what visible here is an aliasing pattern"
- c. in Supplementary Fig. 3 b: "The error bar correspond to"
- d. in Supplementary Note 2B: "We emphasize that the factor 2 in front $\Delta\epsilon_{ij}$ " and "shuttling between QD2 and QD₃,"
- e. in Supplementary Note 8B: "Since, our experimental results"
- f. in Supplementary Note 9: "the spin does not experienced"

We thank the reviewer for reading our manuscript so carefully and pointing out these errors. We corrected them.

Reviewer #2 (Remarks to the Author):

The authors have revised the manuscript to a great extent, including changes in the figure, text and supplementary sections. I believe they address my comments and those of the other reviewers satisfactorily. The impact of high-fidelity shuttling of spin qubits and of quantum information in general is expected to be high, and the paper is of good quality and represents an important step in this area. I therefore would now recommend the paper for publication.

We thank the reviewer for recommending our manuscript for publication.

One more typo: Supplementary Note 8 reads “This could possible”.

We corrected this typo.

Reviewer #3 (Remarks to the Author):

We thank the reviewer for their efforts.

Reviewer #4 (Remarks to the Author):

The authors addressed the review comments and I am comfortable recommending the manuscript for publication.

We thank the reviewer for recommending our work for publication.

REVIEWERS' COMMENTS

Reviewer #1 (Remarks to the Author):

The authors have revised the manuscript and addressed the comments satisfactorily. I am now comfortable recommending the paper for publication.

Minor comment: I suspect that dephasing/relaxation rates in the caption of Supplementary Figure 14 have notation errors.

Reviewer #1 (Remarks to the Author):

The authors have revised the manuscript and addressed the comments satisfactorily. I am now comfortable recommending the paper for publication.

We thank the reviewer for their critical reading of the manuscript and for their support regarding the publication of our work.

Minor comment: I suspect that dephasing/relaxation rates in the caption of Supplementary Figure 14 have notation errors.

Indeed, there was a notation error. We corrected it. Moreover, we have done a final proof reading of the manuscript and made some minor changes to improve the readability and clarity of the manuscript.